# Forcing Mechanisms of the Terdiurnal Tide

Friederike Lilienthal[1], Christoph Jacobi[1], and Christoph Geißler[1]

[1]Institute for Meteorology, Universität Leipzig, Stephanstr. 3, 04103 Leipzig, Germany

**Correspondence:** F. Lilienthal (friederike.lilienthal@uni-leipzig.de)

**Abstract.** Using a nonlinear mechanistic global circulation model we analyze the migrating terdiurnal tide in the middle atmosphere with respect to its possible forcing mechanisms, i.e. the absorption of solar radiation in the water vapor and ozone band, nonlinear tidal interactions, and gravity wave-tide interactions. In comparison to the forcing mechanisms of diurnal and semidiurnal tides, these terdiurnal forcings are less well understood and there are contradictory opinions about their respective

relevance. In our simulations we remove the wavenumber 3 pattern for each forcing individually and analyze the remaining tidal wind and temperature fields. We find that the direct solar forcing is dominant and explains most of the migrating terdiurnal tide's amplitude. Nonlinear interactions due to other tides or gravity waves are most important during local winter. Further analyses show that the nonlinear forcings are locally counteracting the solar forcing due to destructive interferences. Therefore, tidal amplitudes can become even larger for simulations with removed nonlinear forcings.

## 1 Introduction

Atmospheric waves such as solar tides play a crucial role in the dynamics of the mesosphere/lower thermosphere (MLT) region. Tides are global-scale oscillations with periods of a solar day ($24\,h$) or its harmonics ($12\,h$, $8\,h$, etc.). They are mainly the result of absorption of solar radiation in the water vapor (troposphere) and ozone (stratosphere) region. Tidal amplitudes grow with increasing height due to the decrease of density and conservation of energy (e.g., Chapman and Lindzen, 1970; Andrews et al.,

1987). In the MLT, tides can reach wind amplitudes comparable to the magnitude of the horizontal mean wind.

Due to the fact that diurnal tides (DTs) and semidiurnal tides (SDTs) usually have larger amplitudes than the harmonics of higher wavenumbers/higher frequencies, they have attracted more attention in the past and are therefore relatively well understood. However, there are observations of terdiurnal tides (TDTs) showing local amplitudes comparable to those of DTs during some months of the year (Cevolani and Bonelli, 1985; Reddi et al., 1993; Thayaparan, 1997; Younger et al., 2002;

Jacobi, 2012). Observations using midlatitude radar measurements show large TDT amplitudes in autumn and early winter (Beldon et al., 2006; Jacobi, 2012). Namboothiri et al. (2004) also obtained slightly larger amplitudes in winter than in summer while Thayaparan (1997) and Jacobi (2012) additionally emphasize the occurrence of TDTs during spring.

Satellite observations have been used to analyze the TDT on a global scale (Smith, 2000; Moudden and Forbes, 2013; Pancheva et al., 2013; Yue et al., 2013). Yue et al. (2013) presented TDT wind amplitudes from the Thermosphere Ionosphere Mesosphere

Energetics and Dynamics (TIMED) Doppler Interferometer (TIDI) of more than $16\,\mathrm{m\,s^{-1}}$ at $50°$ N/S above $100\,km$ with an additional peak in the meridional component at about $82\,km$ between $10$ and $20°$ N. They identified the first symmetric (3,3)

mode (peaking at $8\,\mathrm{K}$ above the equator and at midlatitudes), using temperatures from Sounding of the Atmosphere using Broadband Emission Radiometry (SABER). At an altitude of $90\,\mathrm{km}$, Moudden and Forbes (2013) found the largest amplitudes above the equator during equinoxes ($6-8\,\mathrm{K}$), and also at $60^\circ\,\mathrm{N}$ during May ($7\,\mathrm{K}$) and at $60^\circ\,\mathrm{S}$ during during October ($5\,\mathrm{K}$) using 10 years of SABER temperature data.

Modeling studies of the TDT are mainly concerned with the analysis of forcing mechanisms (Akmaev, 2001; Smith and Ortland, 2001; Huang et al., 2007; Du and Ward, 2010). This was motivated by the idea that TDTs are not only the consequence of diurnal solar heating but are additionally excited by nonlinear interactions between DTs and SDTs (e.g., Glass and Fellous, 1975; Teitelbaum et al., 1989). The theory for these nonlinear interactions has been outlined by Teitelbaum and Vial (1991) and later by Beard et al. (1999). They state that the period of a child wave $P_3$ resulting from nonlinear interaction is linked to

the periods of the parent waves $P_1$ and $P_2$ through $\frac{1}{P_3} = \frac{1}{P_1} + \frac{1}{P_2}$. The same holds for the wavenumbers. If we consider such a pure nonlinear TDT which is only a result of the interaction between DT and SDT, this means that the wavelength relation between these tides must be:

$$\lambda_{TDT} = \frac{\lambda_{DT}\lambda_{SDT}}{\lambda_{DT} + \lambda_{SDT}} \tag{1}$$

where $\lambda_{DT}$, $\lambda_{SDT}$ and $\lambda_{TDT}$ are the vertical wavelengths of the DT, SDT and TDT, respectively. However, it should be noted

that, in a real atmosphere with unknown contributions of different forcings, this criteria is only sufficient but not necessary to prove the existence of nonlinear interactions. For example, the wavelengths created by nonlinear interactions may not be detected if the solar TDT is stronger and is superposed over the nonlinear TDT. For the same reason, a weak correlation between DT/SDT and TDT amplitudes is not necessarily meaningful.

Another possible excitation source is gravity wave-tidal interactions (e.g., Miyahara and Forbes, 1991; Huang et al., 2007).

More recent simulations (Ribstein and Achatz, 2016) show that details of gravity wave-tidal interactions can change if more comprehensive physics is included but their analysis does not include the TDT.

Teitelbaum et al. (1989) performed the first modeling study on the nonlinear forcing of the TDT and they concluded that the nonlinear interactions and the direct solar forcing lead to comparable terdiurnal amplitudes. Smith and Ortland (2001) used a nonlinear model with specified DT and SDT fields at the lower boundary. They switched off the terdiurnal solar component

on the one hand and removed the direct solar forcing of SDTs on the other hand. As a result, they found that the solar forcing is dominant at middle and high latitudes while nonlinear interactions mainly contribute at low latitudes. A similar approach was applied by Akmaev (2001). They stated that the heating due to absorption of solar radiation in the ozone region is the main source for TDTs, while a noticeable nonlinear contribution is only seen during equinoxes. Huang et al. (2007) used a fully nonlinear tidal model with specified diurnal and semidiurnal thermotidal heating. In this model, the occurrence of

TDT amplitudes was only possible due to nonlinear interactions, and they were significant in the MLT. Another model study about TDT forcing mechanisms was performed by Du and Ward (2010). They analyzed model output from the extended Canadian Middle Atmosphere Model (CMAM) with self-consistent tides due to radiative heating, convective processes and latent heat release. They performed a correlation analysis of DTs and SDTs with TDTs on a seasonal and short-term scale. They concluded that nonlinear interactions are unlikely to be the source of the migrating TDT and that solar heating is the

major source. However, Du and Ward (2010) do not exclude the possibility of nonlinear interactions. They suggest a Hough mode decomposition of the TDT, similar to the analysis of Smith and Ortland (2001). This procedure allows the conclusion to which degree a local forcing actually results in a propagating tidal mode.

To summarize, there are only few modeling studies which address the forcing mechanisms of TDTs, and they do not provide a
consistent perspective. Nonlinear interactions seem to play a tangible role in TDT forcing but to what extent is heavily under debate. To shed more light on this matter we have used a nonlinear global circulation model to explore this issue. To this end we performed model simulations with simultaneous nonlinear and solar terdiurnal forcing. Additional model experiments were undertaken, each with one of the forcing mechanisms switched off, in order to analyze TDT amplitudes due to each forcing, separately.

The paper is arranged as follows: The model and the numerical experiments are described in Sect. 2. Section 3 presents the results of the simulations, starting with an overview on the climatology of the reference TDT in the model. The second part of this section describes the TDTs that are obtained when certain forcings are removed. Finally, in Sect. 4 the results from Sect. 3 are discussed and summarized.

## 2    Description of the Model and the Experiments

We use the nonlinear Middle and Upper Atmosphere Model (MUAM) to investigate the forcing mechanisms of tides with wavenumber 3. MUAM is a 3-dimensional mechanistic model based on the COMMA-LIM (Cologne Model of the Middle Atmosphere – Leipzig Institute for Meteorology) model, which is described in detail by Fröhlich et al. (2003a, b). The more recent version of the model, MUAM, is documented by Pogoreltsev et al. (2007); Pogoreltsev (2007) and Lilienthal et al. (2017). MUAM extends from the surface (1000 hPa) to the lower thermosphere while the zonal mean temperatures in the lower
30 km (i.e. at the lower boundary and 10 height levels above) are nudged towards monthly mean ERA-Interim reanalyses of zonal mean temperature (ERA-Interim, 2018; Dee et al., 2011). Note that this only influences the zonal mean, while waves can still develop unaffected by the nudging. The background winds can freely develop in the model and are only indirectly influenced via the zonal mean temperature nudging. In the present version, there is no additional lower boundary forcing. We perform ensemble simulations for each experiment by using 11 different years (2000-2010) of monthly mean reanalysis input
data, e.g. our results for January are the average of 11 simulations, nudged with 11 different years of January reanalysis data. In contrast to MUAM model experiments performed by Pogoreltsev et al. (2007) or by Jacobi et al. (2015), stationary planetary waves at the lower boundary are not explicitly forced for these model experiments in order to avoid coupling between stationary planetary waves and tides. This is important because an additional secondary coupling with planetary waves leads to a more complex situation with a more complicated quantification of the individual forcing effects.
The model has a horizontal resolution of $5 \times 5.625°$ and a vertical resolution of $2.842$ km in logarithmic pressure height with a constant scale height of $H = 7$ km.

Gravity waves are calculated by an updated Lindzen-type parameterization (Lindzen, 1981; Jakobs et al., 1986) as described by Fröhlich et al. (2003b) and Jacobi et al. (2006). Due to the fact that this parameterization does not account for ionospheric

effects, it is coupled with a modified parameterization after Yiğit et al. (2008), connected via the eddy diffusion coefficient which is calculated in the Lindzen scheme and then transferred to the Yiğit scheme. Gravity waves with phase speeds of 5 to $30\,\mathrm{m\,s^{-1}}$ are handled by the linear Lindzen-type scheme while the Yiğit scheme is restricted to phase speeds of 35 to $105\,\mathrm{m\,s^{-1}}$. Therefore, the intrinsic phase speeds of the waves in the Yiğit scheme are larger than those in the Lindzen-type scheme, so that

they reach their breaking levels at higher altitudes where the amplitude is larger. As a result, the Lindzen-type parameterization essentially affects the stratosphere and mesosphere and the Yiğit parameterization mainly takes effect in the thermosphere. Overlaps between both parameterizations are small and the forcing terms due to gravity waves are summed in the tendency equation of the model. Further parameterizations of solar and infrared radiation as well as several ionospheric effects such as Rayleigh friction, Lorentz force and ion drag are included.

MUAM experiments analyzing TDTs have been performed by Fytterer et al. (2014) who compared the simulated TDT wind shear with global lower ionospheric sporadic E occurrence rates. Additionally, Krug et al. (2015) presented a seasonal climatology of the migrating TDTs based on MUAM simulations.

In the configuration used here, the model incorporates a spin-up of 120 model days. Within that time, zonal mean heating rates (no tides) are building up a background climatology. In the subsequent 90 model days, heating rates are allowed to be zonally

variable and tides start to propagate, gradually increasing in time. The heating rates are fully introduced after model day 154. In this model version, the sun's zenith angle does not account for day to day variations and refers to the first day of the respective month. The solar elevation angle, however, includes a diurnal cycle to account for tidal forcing. The last 30 model days are analyzed and presented here. They represent the mean state of the respective months with an equilibrium of background winds and temperature. Tidal amplitudes remain almost constant and show only small day-to-day variations. Note that the nudging in

the troposphere/lower stratosphere is still active during that period and the model is not running completely freely at any time. This, however, does not influence the tides because the nudging only influences the zonal mean temperature. The background climatology for zonal wind, meridional wind and temperature during solstice (January) and equinox (April) conditions is given in Fig. 1 (for details see section 3.1). This simulation does not include any modifications of the tides and therefore serves as a reference, named REF in the following (see also Table 1).

Within the model there are three mechanisms that may excite TDTs: solar heating, nonlinear interactions between tides and gravity wave-tidal interactions. The first, the diurnal variation of solar heating rates, creates atmospheric tides self-consistently. This mechanism is known to be the most important factor for the forcing of DTs and SDTs (e.g., Andrews et al., 1987). The second mechanism is related to nonlinear interactions between different tides. Following Beard et al. (1999), the interaction between a DT and a SDT can lead to the forcing of a TDT. The last source included in MUAM are gravity waves. Miyahara and

Forbes (1991) have shown that an interaction between gravity waves and the DT can excite a TDT. Trinh et al. (2018) observed a longitudinal variation of gravity wave activity in the tropical MLT region that may also be caused by gravity wave-tidal interaction.

In order to separate these different mechanisms we analyze the wavenumber 3 component of the respective forcing and perform separate model runs in which one of the terdiurnal forcing mechanisms is removed at each model time step for each

latitude/altitude. We do not consider the temporal dimension for this analysis because wavenumber spectra prove that TDTs

in the model are strongest for wavenumber 3 (migrating TDTs) and negligible for other wavenumbers (nonmigrating TDTs, not shown here). This is because nonmigrating tides are usually excited by orographic sources, latent heat release or other geographically fixed effects (e.g., Andrews et al., 1987). Note that atmospheric gases such as water vapor or ozone are only included as zonal means which is different from other versions of MUAM (e.g., Ermakova et al., 2017). Therefore, we usually
refer to the migrating TDT here. The following results are obtained from five ensemble simulations in total, eliminating each forcings separately (NO_SOL, NO_NLIN and NO_GW), allowing all forcings (REF) and eliminating all forcings (CTRL). An overview is given in Table 1.

Note that the background (monthly mean zonal mean) circulation is not significantly altered when TDT forcings are removed (not shown here). Differences amount to not more than the actual standard deviations in the REF simulation (Fig. 1). There-
fore, the influence of a removed wavenumber 3 forcing is comparable to the year-to-year variation of the background state and propagation conditions for tides remain similar.

The parameterization of solar heating in the middle atmosphere is calculated following Strobel (1978). It considers heating due to the most important gases such as water vapor, carbon dioxide, ozone, oxygen and nitrogen. The, zonal mean ozone fields up to $50\,\mathrm{km}$ altitude are taken from the Stratosphere-troposphere Processes And their Role in Climate project (SPARC, 2018;
Randel and Wu, 2007). Above $50\,\mathrm{km}$, the ozone mixing ratio decreases exponentially. The second ozone maximum near $90\,\mathrm{km}$ is not included. In contrast to Jacobi et al. (2015), we restrict our simulations to ozone data of the year 2005 because we do not intend to perform a trend analysis. The volume mixing ratio for carbon dioxide has been chosen according to measurements from Mauna Loa Observatory, also for the year 2005 (e.g., 378ppm for January; NOAA ESRL Global Monitoring Division, 2018; Thoning et al., 1989). Chemical heating due to recombination of $O$ and $O_2$ (Riese et al., 1994) and heating due to ex-
treme ultra violet radiation (EUV) are included. This is described in more detail by Fröhlich et al. (2003a).

In the NO_SOL simulation, the total heating rate of all heating contributions is analyzed using a Fourier transform to separate the tidal components. For the analysis of the forcing mechanism we subtract the wavenumber 3 amplitude from the total heating for each time step and each latitude/altitude, separately. The result of this simulation is a wavenumber 3 tide that is only due to nonlinear interactions and gravity wave effects.
In order to separate the nonlinear forcing we modify the nonlinear terms in the tendency equations of the model (e.g., Jakobs et al., 1986), i.e. in the advection terms in the zonal (Eq. (2)) and meridional (Eq. (3)) momentum equations as well as temperature advection (Eq. (4)):

$$\boldsymbol{v}\cdot(\nabla u) = \frac{u}{a\cos\phi}\frac{\partial u}{\partial\lambda} + \frac{v}{a\cos\phi}\frac{\partial(u\cos\phi)}{\partial\phi} + \frac{w}{\rho_0}\frac{\partial}{\partial z}(\rho_0 u), \tag{2}$$

$$\boldsymbol{v}\cdot(\nabla v) = \frac{u}{a\cos\phi}\frac{\partial v}{\partial\lambda} + \frac{v}{a\cos\phi}\frac{\partial(v\cos\phi)}{\partial\phi} + \frac{w}{\rho_0}\frac{\partial}{\partial z}(\rho_0 v), \tag{3}$$

$$\boldsymbol{v}\cdot(\nabla T) = \frac{u}{a\cos\phi}\frac{\partial T}{\partial\lambda} + \frac{v}{a\cos\phi}\frac{\partial(T\cos\phi)}{\partial\phi} + \frac{w}{\rho_0}\frac{\partial}{\partial z}(\rho_0 T) \tag{4}$$

where $\boldsymbol{v}$ is the wind vector, $u$ and $v$ are the horizontal wind components, $w$ is the vertical wind component and $T$ is the

temperature. $a$ is Earth's radius, $\phi$, $\lambda$ and $z$ are latitude, longitude and altitude, respectively, and $\rho_0$ is the reference density at a given height $z$. Additionally, the adiabatic contribution included in the temperature equation in principle has to be taken into consideration because it includes nonlinear coupling:

$$\left.\frac{\partial T}{\partial t}\right|_{adiab} = \frac{RwT}{m'c_{\mathrm{p}}H} \quad ,$$ (5)

with $R$ as the gas constant for dry air, $m'$ the ratio of molecular weights at the respective altitude and at $1000\,\mathrm{hPa}$ and $c_p$ is the specific heat at constant pressure.

Linearizing these equations, i.e. $T \approx \overline{T} + T'$, $w \approx \overline{w} + w'$, etc., results in a separation of purely nonlinear (wave-wave) interactions, wave-background interactions and pure background processes. For example, the adiabatic term from Eq. (5) may be

written as

$$\left.\frac{\partial T}{\partial t}\right|_{adiab} \approx \frac{R}{m'c_{\mathrm{p}}H} \cdot \left(\overline{w}\,\overline{T} + \overline{w}T' + w'\overline{T} + w'T'\right)$$ (6)

and the terms on the right-hand side of Eqs. 2-4 are treated similarly. The last term in the bracket of Eq. (6) describes nonlinear wave-wave interaction. From these terms of wave-wave interactions we removed the $k = 3$ amplitudes analogous to the modi-

fication of the solar heating terms in the NO_SOL simulation. Removing the nonlinear interactions will result in a combination of solar and gravity wave driven TDT (Run NO_NLIN).

The simulations NO_SOL and NO_NLIN are very similar to the approach presented by Akmaev (2001) and Smith and Ortland (2001). Additionally, we consider gravity waves for the generation of TDTs. The contributions of both gravity wave routines (the Lindzen-type and the modified Yiğit parameterization) to the tendency terms can be simply summed up. The total acceler-

ation of the mean flow due to gravity waves is finally subject to a Fourier filtering of wavenumber 3, similar to the one for the heating rates and the nonlinear terms. As a result, TDTs of solar and nonlinear origin are remaining (NO_GW simulation).

As a control simulation (CTRL), the wavenumber 3 component of the solar, nonlinear and gravity wave forcings are removed simultaneously. This is done in order to test the degree to which all sources of TDTs are captured, and whether the model produces further TDTs of either numerical or physical origin.

In the following analysis, we focus on the months January and April to show solstice and equinox conditions. During this time, the TDT in MUAM is most prominent. Results for July and October are similar and therefore they are not shown, here.

## 3 Results

### 3.1 Reference Simulation: TDT Climatology

The REF simulation includes solar, nonlinear and gravity wave forcing for all wavenumbers. Therefore, it serves as a reference for all the experiments. The following results are given as a mean of the 11 ensemble members, owing to the nudging of re-analysis data for the years 2000-2010 (color shading) with the respective standard deviations (contour lines).

In Fig. 1 we provide a background climatology of the MUAM zonal mean circulation for solstice (Fig. 1a-c) and equinox (Fig. 1d-f) for the parameters zonal wind (a,d), meridional wind (b,e) and temperature (c,f). The color coding denotes the 11-year means, while the standard deviations are given as black contour lines.

Comparing the MUAM climatology with empirical climatologies such as CIRA86 (Fleming et al., 1990), the radar based GEWM (Portnyagin et al., 2004) or the satellite based UARS (Swinbank and Ortland, 2003) we find good agreement but with slightly larger westerly jets and weaker easterly jets during January in MUAM.

We notice that the model produces small year-to-year variations below $100\,\mathrm{km}$ in the southern hemisphere and south of $30°\,\mathrm{N}$. There, the standard deviation $\sigma$ is very small, mostly below $\sigma(T) = 1\,\mathrm{K}$ ($\sigma(u) = 2\,\mathrm{m\,s}^{-1}$, $\sigma(v) = 0.25\,\mathrm{m\,s}^{-1}$). Model variations for middle and high latitudes in the northern hemisphere are larger with standard deviations up to $\sigma(T) = 6\,\mathrm{K}$ ($\sigma(u) = 12\,\mathrm{m\,s}^{-1}$, $\sigma(v) = 2\,\mathrm{m\,s}^{-1}$) during January and $\sigma(T) = 2\,\mathrm{K}$ ($\sigma(u) = 2\,\mathrm{m\,s}^{-1}$, $\sigma(v) = 0.75\,\mathrm{m\,s}^{-1}$) for April. Maxima of the standard deviation are located at about $60°\,\mathrm{N}$. These variations have their origin in the year-to-year variability of the polar vortex for which a range of several K, especially during winter, is realistic. Due to the fact that MUAM assimilates the zonal mean temperature up to $30\,\mathrm{km}$ altitude, this model variability represents a realistic atmospheric variability, too.

Figures 2 and 3 show the terdiurnal component of all forcing terms that our analysis takes into account, namely solar forcing, nonlinear forcing and forcing due to gravity wave-tide interactions. All forcing terms are scaled by $\exp\{-z(2H)^{-1}\}$. This factor is associated with the conservation of wave energy which normalizes the wave growth with height due to the decrease in density. Therefore, the figures show the source region of tidal excitation but they do not provide any information about propagation conditions.

Figure 2 refers to thermal parameters including temperature advection (a,b), the nonlinear component of adiabatic heating (c,d), heating due to gravity waves (e,f) and direct solar heating (g,h). Note that the color scales in Fig. 2 are equal but not continuous in order to cover the magnitudes of all forcings while keeping them comparable to each other. For the thermal forcing of the TDT it can be seen that the direct solar forcing dominates in the troposphere and stratosphere. This is because of the strong absorption of solar radiation by tropospheric water vapor and stratospheric ozone. In the mesosphere (80-100 km), nonlinear effects are mainly responsible for the forcing of terdiurnal fluctuations. Due to absorption of EUV radiation, there is again some solar forcing in the lower thermosphere (Fig. 2g,h at about $120\,\mathrm{km}$ altitude) that is comparable to nonlinear thermal forcing (Fig. 2a,b). In this region, heating due to gravity wave effects (Fig. 2e,f) plays a major role. The nonlinear adiabatic heating effect (Fig. 2c,d) is weak everywhere compared to the other forcings and will therefore be neglected in our further considerations. Figure 3 is similar to Fig. 2 but refers to wind parameters. These are also scaled by the amplitude's growth rate $\exp\{-z(2H)^{-1}\}$. The figure shows nonlinear zonal (a,b) and meridional (c,d) wind advection as well as zonal (e,f) and meridional (g,h) acceler-

ation due to gravity waves. In the zonal wind, in the troposphere and stratosphere, the nonlinear forcing is clearly dominating over gravity wave effects. Zonal gravity wave forcing becomes strong above $100 \, \text{km}$. In January, there is an additional maximum of gravity wave induced terdiurnal forcing (Fig. 3e) near $80 \, \text{km}$ between 30 and $60° \, \text{N}$ which cannot be observed in April (Fig. 3f). For meridional wind patterns, gravity wave forcing only plays a role between $80$ and $100 \, \text{km}$ (Fig. 3g,h), its magnitude being comparable to those of the advective nonlinear forcing (Fig. 3c,d). In the stratosphere and mesosphere, nonlinear advection is the most important source for the meridional component.

Generally, direct solar forcing is weaker during April (Fig. 2h) than during January (Fig. 2g), but most nonlinear forcings (Fig. 2a,b and Fig. 3a-d) become stronger in April and are therefore more dominant during equinox.

As described above, the nonlinear terdiurnal forcing is a result of interactions between the migrating DT and the migrating SDT. These interactions can only take place if both, DT and SDT, have a considerable amplitude. To test this relation between the different harmonics, the product of DT and SDT amplitudes serves as a proxy for the terdiurnal nonlinear forcing. Due to the fact that the forcing terms in Figure 3 are scaled by the growth rate of the amplitudes, we also scaled the product of DT and SDT amplitudes to show the source region of the possible nonlinear interaction. As an example, Fig. 4 shows the results for temperature (a), zonal wind (b) and meridional wind amplitudes (c) during January.

It can be seen that the scaled product of DT and SDT amplitudes exhibits similar structures to the nonlinear terdiurnal forcing terms in Fig. 2a and Fig. 3a,c. For example, the zonal and meridional component (Fig. 4b,c), have regions of enhanced amplitude near $50 \, \text{km}$ extending from low latitudes poleward to high latitudes and with a minimum over the equator. This is in good agreement with the nonlinear zonal and meridional forcing in Fig. 3a,c. The similarities for the temperature component (Fig. 4a and Fig. 2a) are less clear but we want to emphasize that the multiplied amplitudes of DT and SDT only serve as proxy. The pure existence of an overlapping DT and SDT source region does not necessarily induce an interaction.

TDT amplitudes are presented for January (Fig. 5a-c) and April (Fig. 5d-f). In contrast to the forcing terms, they are not scaled. Zonal wind amplitudes become stronger in April (Fig. 5e) compared to January (Fig. 5b) above $110 \, \text{km}$ but this is not the case for the temperature and meridional wind amplitude. Amplitudes at $100 \, \text{km}$ altitude reach only about $1.5 \, \text{K}$ and $4 \, \text{m\,s}^{-1}$ (zonal/meridional wind). This is much smaller than observed by radars (e.g., Thayaparan, 1997; Namboothiri et al., 2004; Beldon et al., 2006; Jacobi, 2012) and by satellite measurements (e.g., Moudden and Forbes, 2013; Pancheva et al., 2013; Yue et al., 2013). They reported amplitudes of about $5\text{-}6 \, \text{m\,s}^{-1}$ at $90 \, \text{km}$ (Thayaparan, 1997; Namboothiri et al., 2004) during equinoxes and local winter with a minimum during summer. These radars are located between $40\text{-}50° \, \text{N}$ and in these regions, Fig. 5 also shows larger wind amplitudes during winter and equinoxes. Beldon et al. (2006) and Jacobi (2012) observe a maximum larger than $10 \, \text{m\,s}^{-1}$ ($95 \, \text{km}$) during autumn/early winter and a smaller one during spring. The absence of a mid-winter maximum can be explained by the location of the radars ($> 50° \, \text{N}$) which is north of the region with a winter maximum as can be seen in Fig. 5b,c.

However, considering only the maxima does not give a good comparison between seasons, and the height-latitudinal structure is more important. Especially in temperature (Fig. 5a) and zonal wind (Fig. 5b) we note a double-peak structure in January with maxima at very low latitudes and a minimum at the equator. This turns into a triple-peak structure in April (Fig. 5d,e) with maxima slightly more poleward ($30° \, \text{N/S}$) and directly at the equator. This structure is also visible in SABER measurements

reported by Pancheva et al. (2013) for March and December. In the meridional wind, the structure of the TDT is not as clear in January (Fig. 5c), with several maxima between $\pm 60°$, the strongest one appearing near the equator. In April (Fig. 5f), it has four distinct peaks with maxima at low and midlatitudes but, in contrast to temperature and zonal wind, a minimum at the equator. This pattern of superposed maxima on minima and vice versa between the zonal and meridional wind components is expected from the wave structure itself.

The standard deviation of tidal amplitudes is relatively small, not more than $10\%$ of the total amplitude. Thus, our results prove to be robust in structure and strength.

The TDT phases are shown in Fig. 6. At each latitude, the corresponding vertical wavelength can be obtained from the vertical phase gradient. The wavelength is taken as the vertical distance between two points of identical phases. A full span of phases should be covered between these points, and for upward propagating waves, the phase gradient for the determination should be negative. Where the amplitude is large, vertical wavelengths turn out to be longer, i.e. the vertical phase gradients are small. Where the amplitude is small, wavelengths are shorter with larger phase gradients. Thayaparan (1997), Namboothiri et al. (2004) and Jacobi (2012) report a similar relationship with vertical wavelengths being short in summer when the amplitude minimizes. Typically, the wavelengths in Fig. 6 reach $100\,\mathrm{km}$ and more. In January (Fig. 5a-c), the structure of phases appears to be more complex while in April (Fig. 5d-f) there are large areas of constant phase, especially at low latitudes.

Figure 7 presents the seasonal cycle of TDT amplitudes at an altitude of $106\,\mathrm{km}$. Results of satellite data analyses have frequently been presented at 90 and $110\,\mathrm{km}$ (Pancheva et al., 2013; Yue et al., 2013; Moudden and Forbes, 2013), and therefore we choose an altitude between these heights. The temperature TDT at this altitude (Fig. 7a) appears to be strongest during equinoxes near the equator ($3.0\,\mathrm{K}$) and at midlatitudes ($30\text{-}40°$ N/S). The amplitudes in autumn ($2.2\,\mathrm{K}$) are larger than those in spring ($1.6\,\mathrm{K}$). Further maxima are present during local winter at $30\text{-}40°$ N/S ($2.6\,\mathrm{K}$ at northern hemisphere and $2.3\,\mathrm{K}$ at southern hemisphere). For latitudes poleward of $50°$ N/S, amplitudes are much lower and peak during summer ($< 1.1\,\mathrm{K}$).

The structure of MUAM temperature amplitudes is generally confirmed by SABER measurements (e.g., Moudden and Forbes, 2013; Pancheva et al., 2013; Yue et al., 2013) who reported maxima of about $5\,\mathrm{K}$ during equinoxes near the equator at $90\,\mathrm{km}$ altitude. Note that this amplitude is almost twice as large as the one obtained from our model simulations even though the altitude is smaller. For midlatitudes, Moudden and Forbes (2013) also found maxima during northern winter ($3\text{-}4\,\mathrm{K}$ at $30\text{-}50°$ N, $90\,\mathrm{km}$) but not during southern winter. This is in contrast to the results of Pancheva et al. (2013) and Yue et al. (2013) who found maxima during equinoxes and local winter in both hemispheres ($110\,\mathrm{km}$ altitude) which qualitatively agrees well with our results at $106\,\mathrm{km}$ near $40°$ N/S.

Maxima in zonal wind (Figs. 7b) and meridional wind TDT (Figs. 7c) are also found during local winter at midlatitudes. They are slightly larger in the northern hemisphere ($5.9\,\mathrm{m\,s^{-1}}$ in both components) than in the southern hemisphere ($4.7\,\mathrm{m\,s^{-1}}$ in both components). During equinoxes, the maxima are smaller and located close to the equator (zonal wind only, $< 4.0\,\mathrm{m\,s^{-1}}$), at low latitudes (meridional wind only, $< 4.3\,\mathrm{m\,s^{-1}}$) and at midlatitudes (zonal and meridional wind $< 3.8\,\mathrm{m\,s^{-1}}$).

Zonal and meridional amplitudes at midlatitudes ($40\text{-}50°$ N/S) agree well with TIDI measurements (Yue et al., 2013) showing maxima during equinoxes at both hemisphere and during southern winter. However, the northern winter maximum is not seen in the zonal wind analysis by Yue et al. (2013). Another meridional wind peak is reported by Yue et al. (2013) near $30°$ N

during July which can be found in our simulations, as well. However, amplitudes tend to be underestimated by a factor of about 3-4.

Some differences between model results and satellite measurements may be explained by the orbit of the satellite, passing high latitudes less frequently and leading to larger uncertainties at these latitudes. However, this cannot explain the large discrepancies in the magnitude of the TDT. Smaller model amplitudes may be due to processes that are not included in the simulations such as latent heat release.

## 3.2 Separating the Forcings

In order to determine the effect of each individual forcing on the amplitude of the TDT we performed the simulations with different forcings switched off, as listed in Table 1.

NO_SOL represents a TDT that is only due to nonlinear and gravity wave effects because wavenumber 3 direct solar heating is removed in the whole model domain. Therefore, possible sources of this wave are nonlinear interactions between other tides, i.e. between the DT and the SDT, and gravity wave-tide interactions only. The resulting amplitudes and phases are shown in Figs. 8 and 9. As expected, the amplitudes are strongly reduced. However, they are not completely extinguished. In all parameters there is a clear maximum at northern midlatitudes (about $60°$ N) during January reaching $4\,\text{K} \pm 0.6\,\text{K}$ (temperature), $5\,\text{m}\,\text{s}^{-1} \pm 1.2\,\text{m}\,\text{s}^{-1}$ (zonal wind) and $4\,\text{m}\,\text{s}^{-1} \pm 0.8\,\text{m}\,\text{s}^{-1}$ (meridional wind) above $120\,\text{km}$. In the zonal wind component there is a secondary maximum at about $30°$ N as well. During April, the maxima are shifted towards the equator with amplitudes similar to those in January. This indicates that secondary terdiurnal forcing is most evident during local winter as is confirmed from the annual cycle of the NO_SOL simulation (not shown here). TDT phases from this simulation (Fig. 9) are much more irregular in comparison to the REF simulation (Fig. 6) and show vertical wavelengths shorter than $50\,\text{km}$ for those latitudes where TDT amplitudes are strong.

The simulation NO_NLIN only includes direct solar forcing and gravity wave-tide interactions. Therefore, it does not include nonlinear interactions. Figure 10 shows the mean amplitude differences between the NO_NLIN and REF ensembles where red (blue) colors denote larger (smaller) amplitudes in NO_NLIN. Welch's t-test was applied and areas with $\alpha < 0.01$ are hatched. It turns out that decreased amplitudes are not the only consequence of the removed nonlinear forcing since there are also areas where the amplitude has increased. This result occurs primarily during January in all parameters. The strongest increase of about $3\,\text{K}$ ($3\,\text{m}\,\text{s}^{-1}$) is located where the REF amplitude reaches its maximum. There, the amplitude in the NO_NLIN simulation is about $25\%$ larger compared to the REF simulation. Another large red area is located at about $60°$ N at an altitude of $110\,\text{km}$. In this area the amplitudes in the REF simulation are relatively small (not more than $2\,\text{K}$ and $2\,\text{m}\,\text{s}^{-1}$), but the differences between the simulations are similar so that the NO_NLIN amplitudes are twice as strong as the REF amplitudes. In April only weak enhancements of about $0.5 - 1.5\,\text{K}$ ($0.5 - 2\,\text{m}\,\text{s}^{-1}$) appear for individual grid points and these are not located in the areas of larger amplitudes. Generally, the negative amplitude differences dominate and areas of positive change are negligible.

We do not show the phases of the NO_NLIN simulation and the NO_GW simulation here because both of these simulations

still include the solar forcing which dominates the other remaining forcing. As a result, the phases are almost identical to those shown in Fig. 6 for the REF simulation.

In order to investigate the reason for the positive difference in amplitude it is useful to compare phase shifts $\Delta\phi$ between the NO_NLIN TDT (with solar and gravity wave forcing) and the NO_SOL TDT (with nonlinear and gravity wave forcing).

The gravity wave forcing appears in both simulations and therefore the phase shift between the tides associated with these simulations can be mainly attributed to the phase shift between a pure solar wave and a pure nonlinear wave. The differences in the background wind and therefore tidal propagation conditions between the simulations are small. For $120° < \Delta\phi < 240°$ destructive interference occurs and leads to a decrease in amplitude for the case of superposition.

Figure 11 shows the amplitude differences as presented in Fig. 10 but now scaled by the growth rate (factor $\exp\{-z(2H)^{-1}\}$)

to show the source of the positive amplitude differences. Here, the hatched areas show regions of destructive interference ($120° < \Delta\phi < 240°$) between the phases of NO_NLIN and NO_SOL occur. It is clear that the red areas and the destructive interferences match almost perfectly for both January and April conditions and for all parameters.

Figure 12 shows the mean amplitude differences between the NO_GW and REF ensembles. For this simulation, positive amplitude differences occur at several heights/latitudes, when the gravity wave-tide interaction as a forcing of TDTs is removed.

In this case destructive interference seems to be more independent of the season as can be seen in January and April. However, the regions where the zonal wind amplitude has increased are rather small. This increase is most apparent around $60°$ N and $110\,\mathrm{km}$ altitude during January. Note that this area is positive for both the meridional and zonal wind and also appears in the NO_NLIN simulation (Fig. 10). For the temperature and meridional wind component we find, as in Fig. 10, that amplitudes in regions with strong REF amplitudes are enhanced even when the wavenumber 3 gravity wave forcing is removed. Furthermore,

the amplitude changes in Fig. 12 are larger during April compared to January. This is consistent with the larger TDT reference amplitudes occurring during April. Generally, all amplitude differences are stronger for NO_GW than for NO_NLIN.

The CTRL simulation provides a measure of TDT amplitudes due to effects that have not been considered in the previous simulations. The presence of regions of significant amplitude indicates that there still exist other sources in the model. Figure 13 shows the TDT amplitudes for the CTRL simulation. Note that the scale is different from Fig. 5 so that the smaller magnitudes

can be seen. The structure of this remaining tide is not completely irregular. However, the amplitudes are small with maximum values below $1\,\mathrm{K}$, $1.2\,\mathrm{m\,s^{-1}}$ (zonal wind) and $1.4\,\mathrm{m\,s^{-1}}$ (meridional wind). During January, maxima are located in the northern hemisphere at low and midlatitudes and during April at the equator (temperature) and at southern low and midlatitudes (wind).

## 4 Discussion and Conclusion

The results of our REF simulation present a climatology and structure of the TDT that generally agrees with observations and earlier model studies. MUAM produces relatively small amplitudes for the TDT, e.g. $5\,\mathrm{m\,s^{-1}}$ for the zonal wind component at $106\,\mathrm{km}$ altitude during winter or $12\,\mathrm{m\,s^{-1}}$ at an altitude of $120\,\mathrm{km}$ during April. In fact, it is an ongoing issue that numerical models tend to underestimate tides, at least for some regions or seasons (e.g., Smith, 2012; Pokhotelov et al., 2018).

In contrast to reports by Cevolani and Bonelli (1985); Reddi et al. (1993); Thayaparan (1997) or Yue et al. (2013) the TDT in our simulations does not attain the amplitude of a typical DT or SDT. However, this property of the TDT was mainly reported for short temporal scales of only few days which are not represented in MUAM.

MUAM simulations show strongest wind amplitudes at midlatitudes (30-50° N) during winter with smaller maxima during spring and autumn. This is in accordance with radar measurements at these latitudes (e.g., Thayaparan, 1997; Namboothiri et al., 2004) who observed amplitudes of at least $5\,\mathrm{m\,s^{-1}}$ during the whole year except during summer. At slightly higher latitudes (50-60° N), the winter maxima disappear and those near the equinoxes become more important as reported by (e.g., Beldon et al., 2006; Jacobi, 2012). There are also agreements with satellite analyses by Moudden and Forbes (2013), e.g., during equinoxes maxima appear at the equator and at midlatitudes. However, Moudden and Forbes (2013) observe those maxima to be more poleward (at about 60° N/S) than we do (30-40° N/S in MUAM). They also find that winter maxima are located about 30-40° N while poleward of 55° the maxima appear during summer.

The TDT in model simulations by Smith and Ortland (2001) has wind maxima near 50°N/S in the respective winter hemisphere. This is slightly more equatorward but generally agrees with our results. They have peak amplitudes of about $10\,\mathrm{m\,s^{-1}}$, about twice as large as in MUAM. In our model, the zonal wind amplitudes at low latitudes are generally weaker than at midlatitudes. Slightly enhanced amplitudes can be seen at low latitudes in the summer hemisphere and during equinoxes above the equator. This structure is similar to the TDT reported by Smith and Ortland (2001) at 97 km and by Du and Ward (2010) at 95 km but vanishes at higher altitudes (Du and Ward, 2010). The temperature amplitude in our model has a strong maximum during equinoxes at the equator and at midlatitudes. This is not seen in earlier simulations (Du and Ward, 2010) but agrees with observations (e.g. Beldon et al., 2006; Jacobi, 2012; Moudden and Forbes, 2013).

In order to investigate the different generation mechanisms of the TDT we present their respective source regions. In addition to the methods used by, e.g. Akmaev (2001); Smith and Ortland (2001); Huang et al. (2007) or Du and Ward (2010), who focus on direct solar heating and nonlinear interactions between tides only, we also consider gravity wave-tide interactions as suggested by, e.g. Miyahara and Forbes (1991) and Huang et al. (2007). To summarize, the solar forcing is dominant in the troposphere and stratosphere, nonlinear interactions are present in the mesosphere, and the forcing due to gravity waves is only significant in the zonal component above the mesopause. This analysis, however, only allows insight into the local terdiurnal forcing, which does not necessarily result in a propagating tide. As suggested by Du and Ward (2010), a Hough mode decomposition of the forcing terms and of the actual tide could shed more light into this issue. Here, a different approach has been applied to analyze the propagating tides due to different forcing mechanisms. Similar to Akmaev (2001) and Smith and Ortland (2001), we perform further model simulation where possible forcing mechanisms are switched off individually.

Removing the direct terdiurnal solar heating leads to a significant decrease in amplitude (see Fig. 8) and therefore we conclude that the solar forcing is the most important and dominant TDT source amongst all possible mechanisms. With respect to the relevance of the solar forcing, our results generally agree with earlier simulations by Smith and Ortland (2001); Akmaev (2001) and Du and Ward (2010). However, the amplitudes in our simulations associated with the additional forcing mechanisms amount to several K or $\mathrm{m\,s^{-1}}$, respectively, at few latitudes/altitudes, which is about one third to one half of the total amplitude. This gives rise to the assumption that nonlinear interaction between tides and/or gravity waves should also be considered as an

important forcing. The „left-over amplitudes", which include nonlinear and gravity wave induced forcing, exhibit a maximum at northern low and midlatitudes during January and April alike and phases for this tide are much more complex than those associated with solar heating (Figs. 8 and 9). Huang et al. (2007) also underline the importance of nonlinear interactions but they obtain pure nonlinear tidal amplitudes up to $15\,\mathrm{m\,s^{-1}}$ and $12\,\mathrm{K}$ during equinoxes near $100\,\mathrm{km}$. These amplitudes are much
larger than those in MUAM. In contrast, the simulations by Smith and Ortland (2001) reveal that nonlinear interactions are weak and only contribute at low latitudes.

Removing the nonlinear tidal interactions leads to an increase in amplitude for some heights/latitudes during January by up to $2\,\mathrm{K}$ ($3\,\mathrm{m\,s^{-1}}$). Although Smith and Ortland (2001) and Akmaev (2001) used the same procedure to analyze the solar and nonlinear forcing contribution, they did not observe this behavior of increased amplitudes. However, Smith et al. (2004) studied
the forcing mechanisms of the quarterdiurnal tide (period of $6\,\mathrm{h}$) and they have seen a similar feature. They conclude that the nonlinear forcing may reduce rather than enhance the tide. This can be explained as a result of destructive interferences between the purely solar forced tide and the nonlinearly forced tide. Due to the destructive phase shift the waves are counteracting each other and therefore reduce the amplitude when appearing together.

Similar results are obtained when the terdiurnal gravity wave-tide interactions are removed but an increase in amplitude in
this case is observed for both January and April conditions. Here, the zonal wind component is not affected by this positive amplitude change but temperature and meridional wind are.

This conclusion supports the results of Smith and Ortland (2001) and partly those of Akmaev (2001) who found some minor nonlinear contributions but assume the solar forcing to be a major source. While Smith and Ortland (2001) also obtain largest nonlinear contribution at low and middle latitudes, Akmaev (2001) point out that nonlinear interactions take place during
equinoxes. However, Akmaev (2001) only analyzed a latitude of $44°\,\mathrm{N}$ where amplitudes seem to maximize during equinoxes and therefore one may conclude that nonlinear interactions generally come into play where the TDT is large. Therefore, we cannot agree with Du and Ward (2010) who concluded that nonlinear interactions are negligible. However, we did not perform a correlation analysis between DTs, SDTs and TDTs and therefore we cannot directly compare with their results. Furthermore, our simulations do not agree with Huang et al. (2007) who obtain very large wind amplitudes over $15\,\mathrm{m\,s^{-1}}$ and temperature
amplitudes over $10\,\mathrm{K}$ in the MLT region for TDTs due to nonlinear interactions only. However, they also find nonlinear amplitude maxima during equinoxes at low and middle latitudes which is in agreement with our results.

Finally, a control simulation (CTRL) tested the TDT amplitude when all three forcings considered here are removed simultaneously to check whether there is a remaining weak forcing that has not yet been considered. Amplitudes for that simulation are relatively small ($< 0.6\,\mathrm{K}$ and $< 1.5\,\mathrm{m\,s^{-1}}$) but have a clear structure with maxima at $50°\,\mathrm{N/S}$ during local winter. Rind et al.
(2014) have noted that numerical noise can produce regular signatures like a quasi-biennial oscillation. Therefore, noise cannot be excluded as a tidal source in the CTRL simulation. Another reasonable TDT source in our model could be originating from the thermospheric parameterizations, which include some nonlinear terms. These sources, however, are likely to be dependent on the model used and it is not likely that the remaining amplitudes in Fig. 13 have a real meteorological meaning.

In the future, it would be interesting to analyze nonmigrating tides, as well. To do this, we would need to include additional
sources such as latent heat release or 3-dimensional ozone and water vapor (e.g., Ermakova et al., 2017). As we have seen,

gravity waves are a crucial parameter for tidal forcing and they also have a large influence on the background circulation of the middle atmosphere. Therefore, the coupling of two different gravity wave parameterization is going to be replaced by an original whole atmosphere scheme after Yiğit et al. (2008).

5 *Code availability.* The MUAM model code can be obtained from the corresponding author on request.

*Author contributions.* F. Lilienthal designed and performed the MUAM model runs. C. Jacobi together with F. Lilienthal drafted the first version of the text. C. Jacobi and C. Geißler contributed to the analysis and interpretation of the results.

*Competing interests.* We declare that no competing interests are present.

*Acknowledgements.* This research has been funded by Deutsche Forschungsgemeinschaft under grant JA 836/30-1. SPARC global ozone
10 fields were provided by W.J. Randel (NCAR) through ftp://sparc-ftp1.ceda.ac.uk/sparc/ref_clim/randel/o3data/. Mauna Loa carbon dioxide mixing ratios were provided by NOAA through ftp://aftp.cmdl.noaa.gov/data/trace_gases/co2/flask/surface/. ERA-Interim data have been provided by ECMWF on http://apps.ecmwf.int/datasets/data/interim_full_moda/?levtype1/4pl.
We further acknowledge support from the German Research Foundation (DFG) and Universität Leipzig within the program of Open Access Publishing.

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

**Table 1.** Overview on the different simulations.

| Simulation | Description | solar forcing | nonlinear forcing | gravity wave forcing |
|---|---|---|---|---|
| REF | Reference with all forcings | on | on | on |
| NO_NLIN | Effect of removed nonlinear forcing | on | off | on |
| NO_SOL | Effect of removed solar forcing | off | on | on |
| NO_GW | Effect of removed gravity wave forcing | on | on | off |
| CTRL | Control without all forcings | off | off | off |

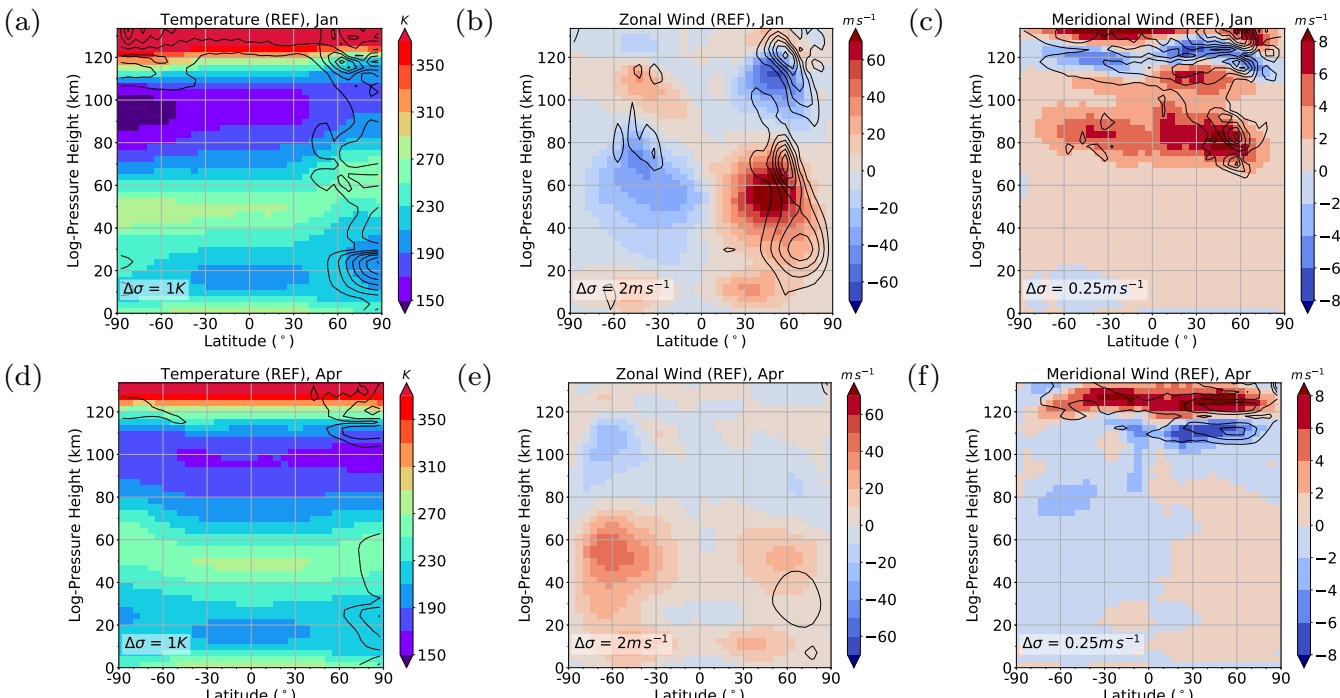

**Figure 1.** From left to right: REF zonal mean temperature, zonal wind and meridional wind. Top: Solstice (January) conditions. Bottom: Equinox (April) conditions. Results are an average of the 11 ensemble members (color shading). Standard deviations $\sigma$ are added as black contour lines and intervals $\Delta\sigma$ are given in each panel.

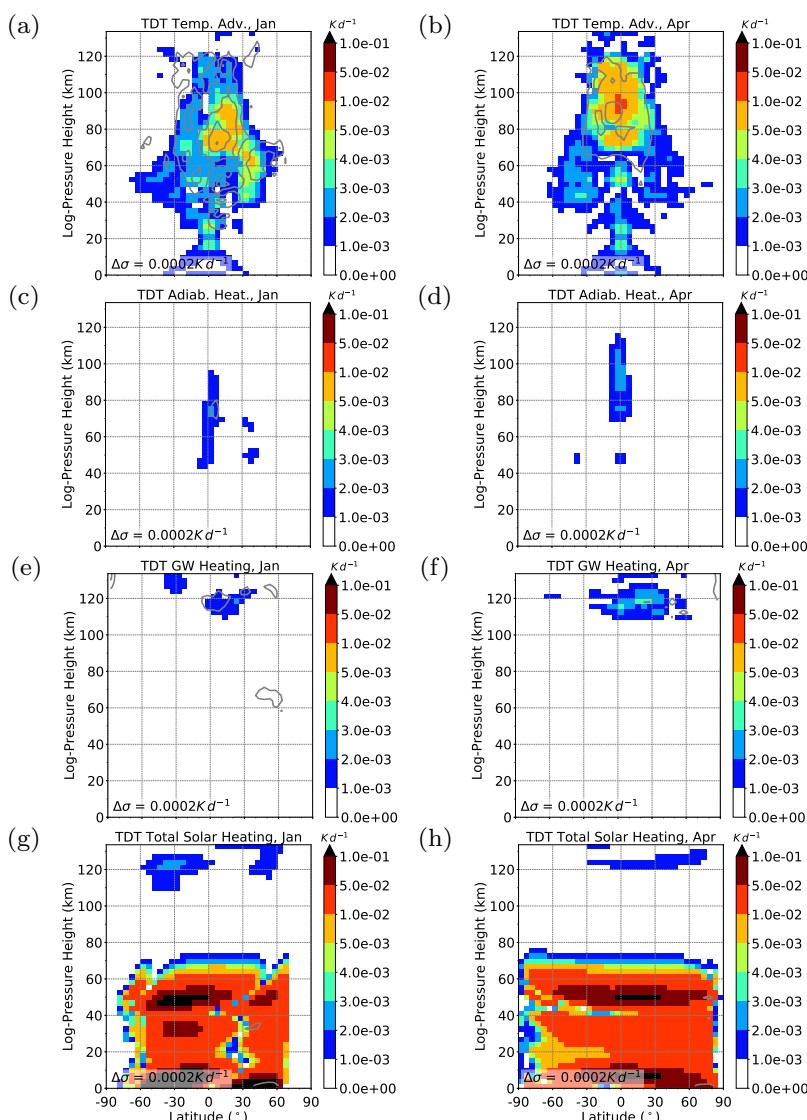

**Figure 2.** Terdiurnal component of thermal tendency terms in the REF simulation for January conditions (left) and April conditions (right). Amplitudes are scaled by $\exp\{-z(2H)^{-1}\}$. Results are an average of the 11 ensemble members (color shading). Standard deviations ($\sigma$) are added as gray contour lines. From top to bottom: temperature advection (nonlinear component of Eq. (4)), adiabatic heating (nonlinear component of Eq. (5)), heating due to gravity wave activity (tendency term from gravity wave parameterization) and solar heating (tendency term from solar radiation parameterization). Note that the color scale is not linear.

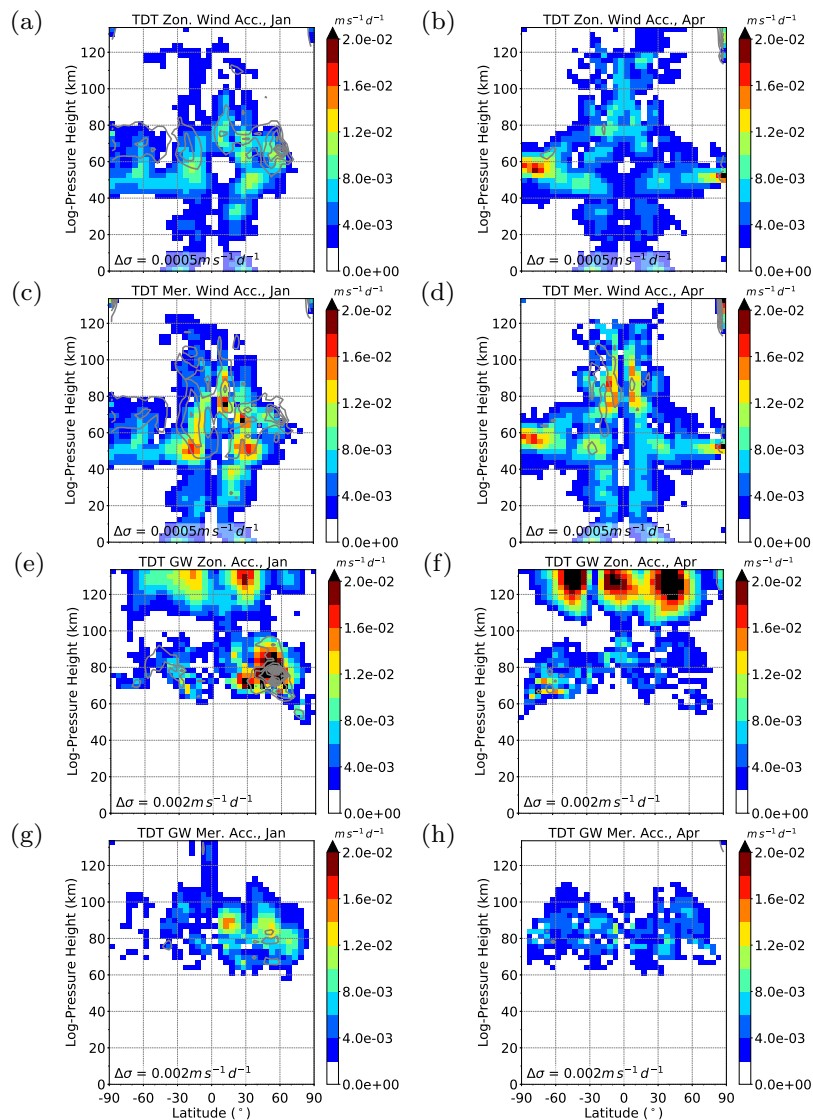

**Figure 3.** Terdiurnal component of zonal and meridional wind acceleration terms in the REF simulation for January conditions (left) and April conditions (right). Amplitudes are scaled by $\exp\{-z(2H)^{-1}\}$. Results are an average of the 11 ensemble members (color shading). Standard deviations ($\sigma$) are added as gray contour line. From top to bottom: zonal wind advection (nonlinear component of Eq. (2)), meridional wind advection (nonlinear component of Eq. (3)) and zonal and meridional acceleration due to gravity waves (tendency terms from gravity wave parameterization).

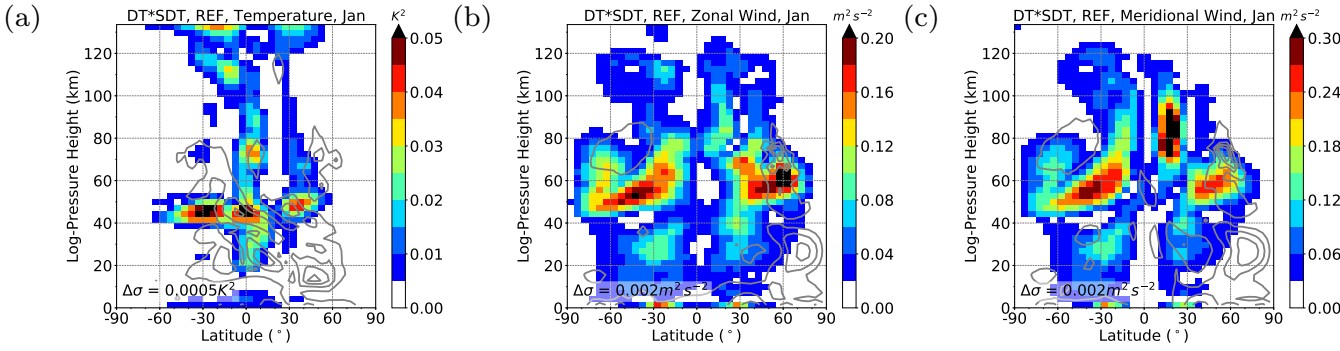

**Figure 4.** Product of DT and SDT amplitudes, scaled by $\exp\{-z(2H)^{-1}\}$ for temperature (a), zonal wind (b) and meridional wind (c); January conditions. Results are an average of the 11 ensemble members (color shading). Standard deviations ($\sigma$) are added as gray contour lines.

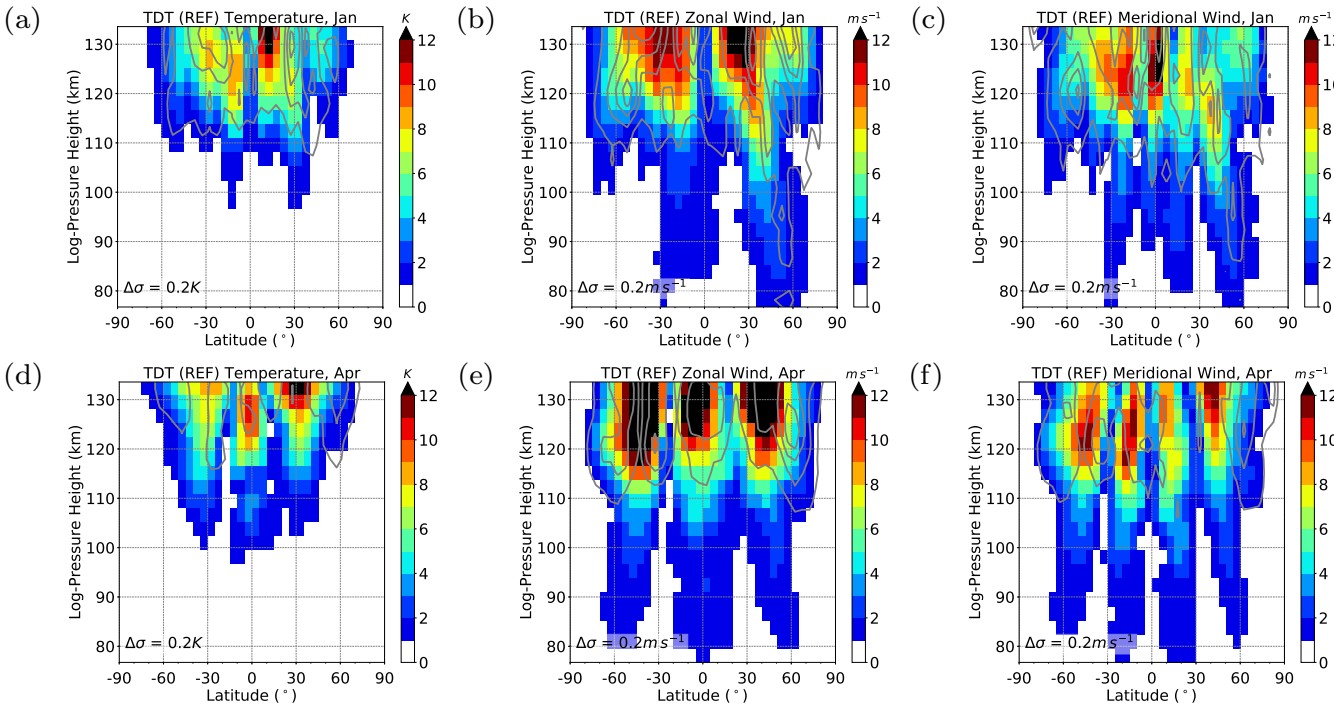

**Figure 5.** Zonal mean TDT amplitudes (colors, REF). From left to right: Temperature, zonal wind, meridional wind. Top: Solstice (January) conditions. Bottom: Equinox (April) conditions. Standard deviation ($\sigma$) are added as gray contour lines.

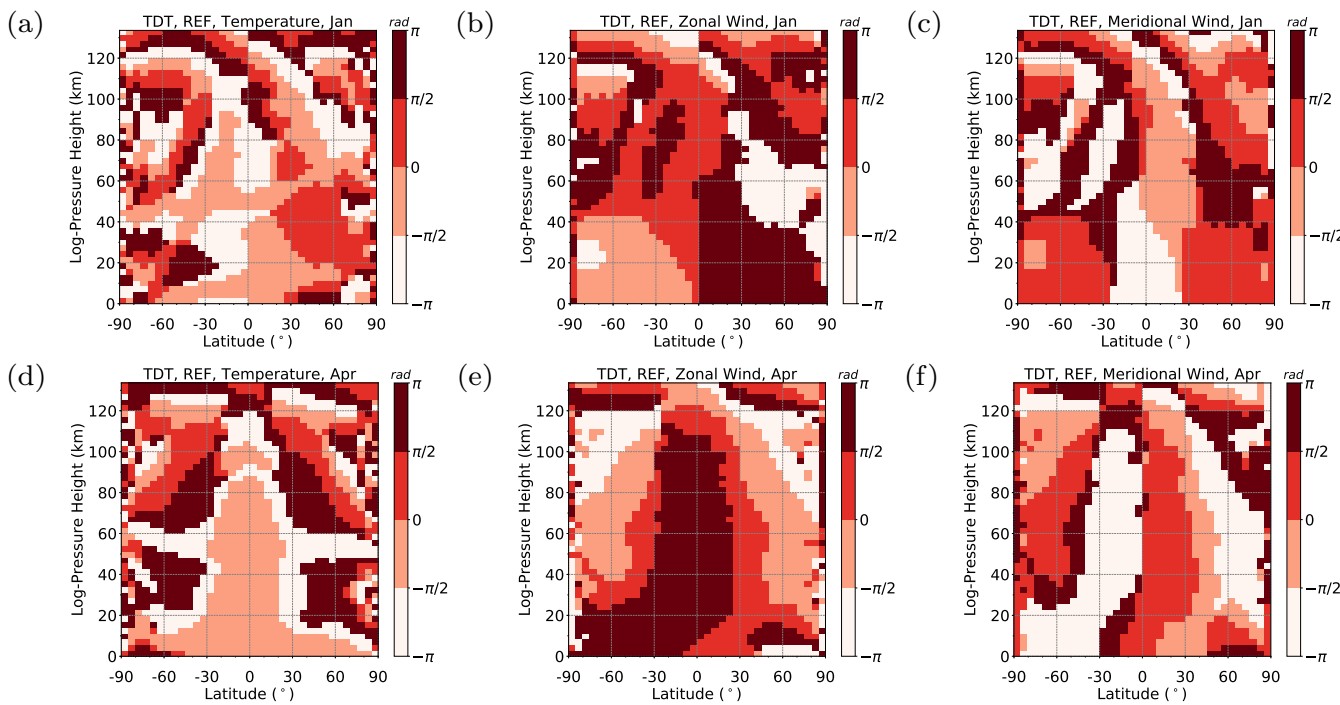

**Figure 6.** Zonal mean TDT phases (REF). From left to right: Temperature, zonal wind, meridional wind. Top: Solstice (January) conditions. Bottom: Equinox (April) conditions.

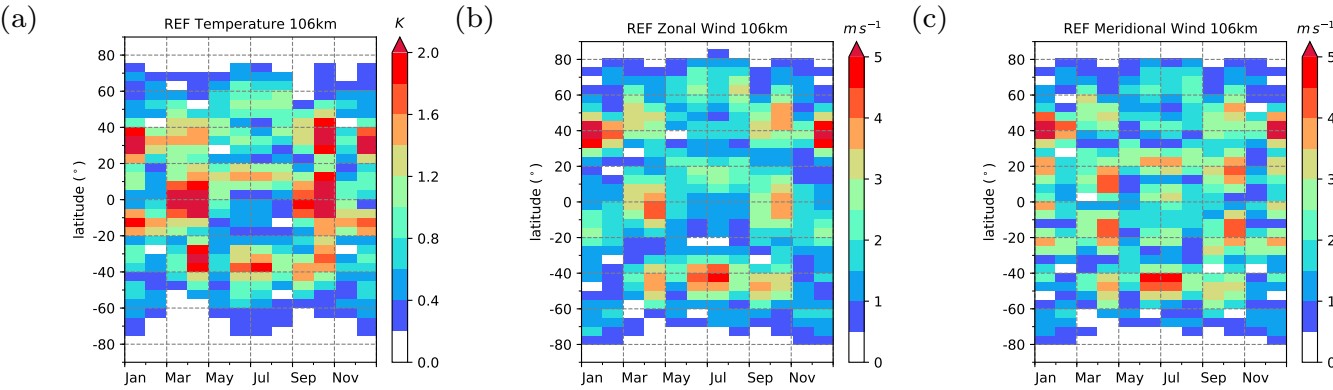

**Figure 7.** REF monthly mean TDT amplitudes at an altitude of $\approx 106$ km. From left to right: temperature, zonal wind component, meridional wind component.

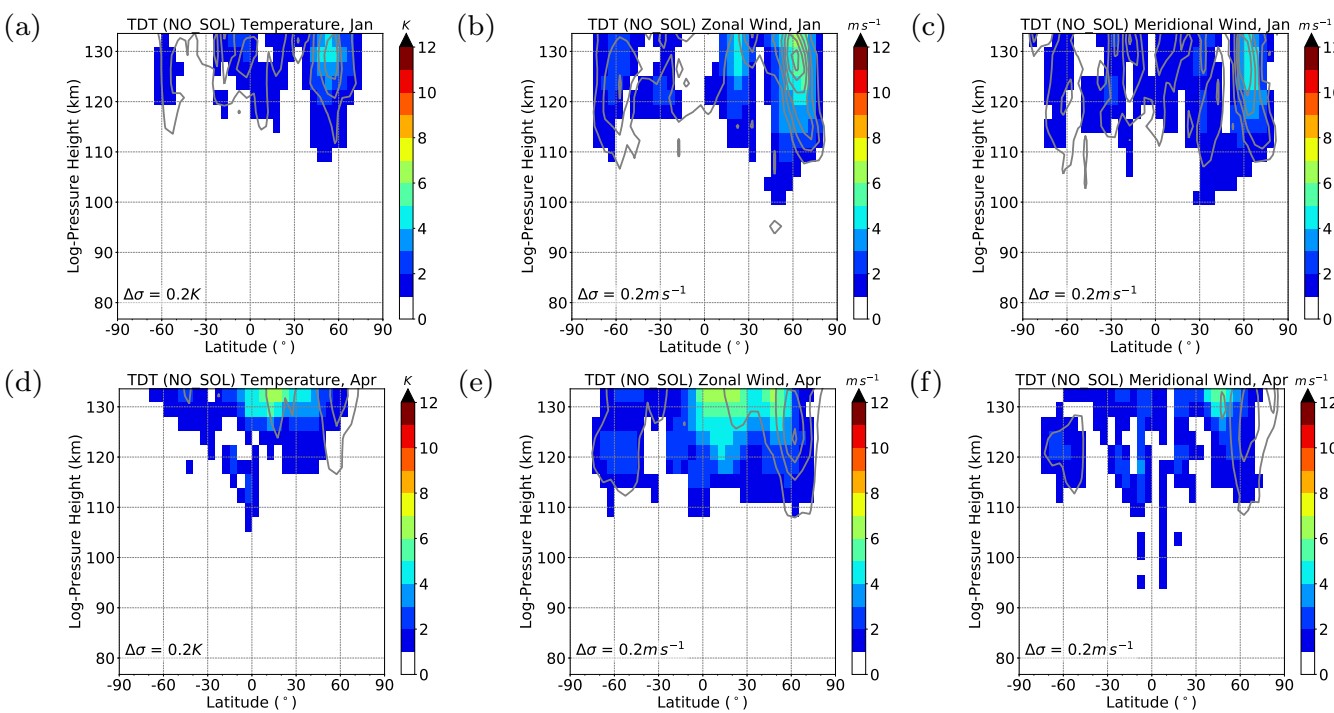

**Figure 8.** As in Fig. 5 but for NO_SOL simulation.

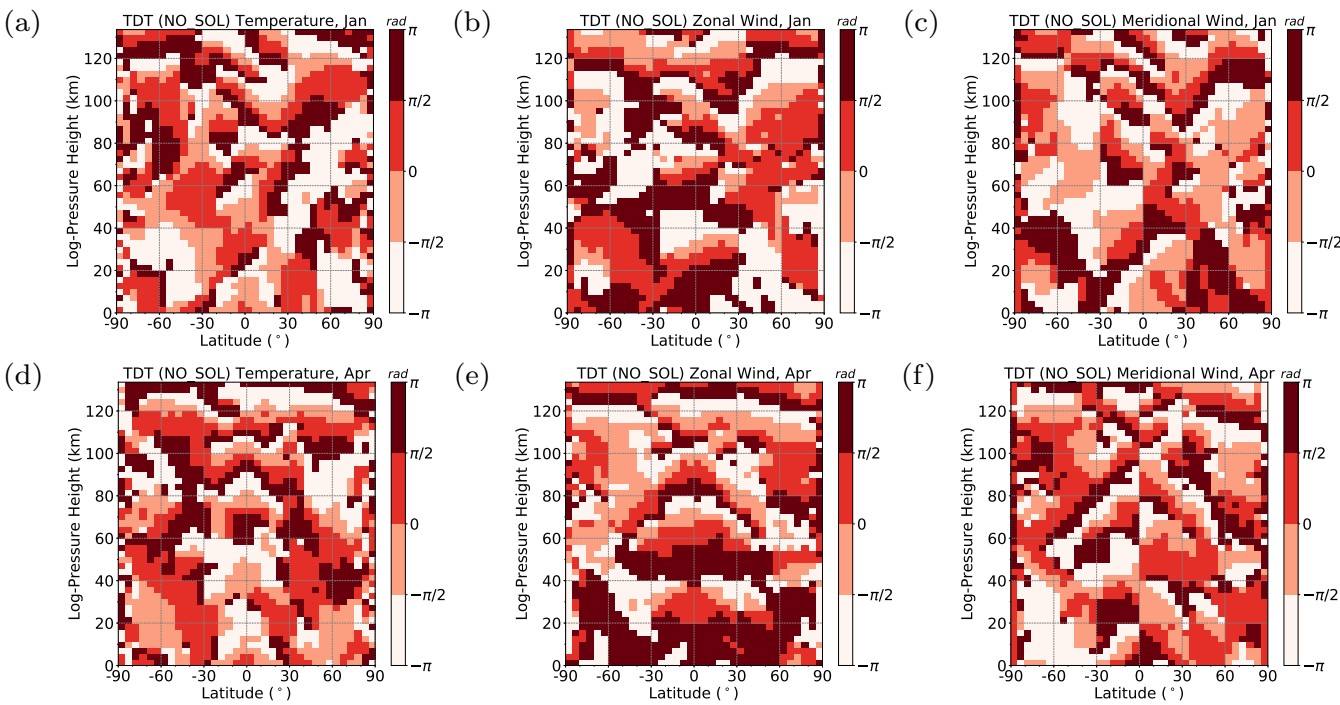

**Figure 9.** Zonal mean TDT phases (NO_SOL). From left to right: Temperature, zonal wind, meridional wind. Top: Solstice (January) conditions. Bottom: Equinox (April) conditions.

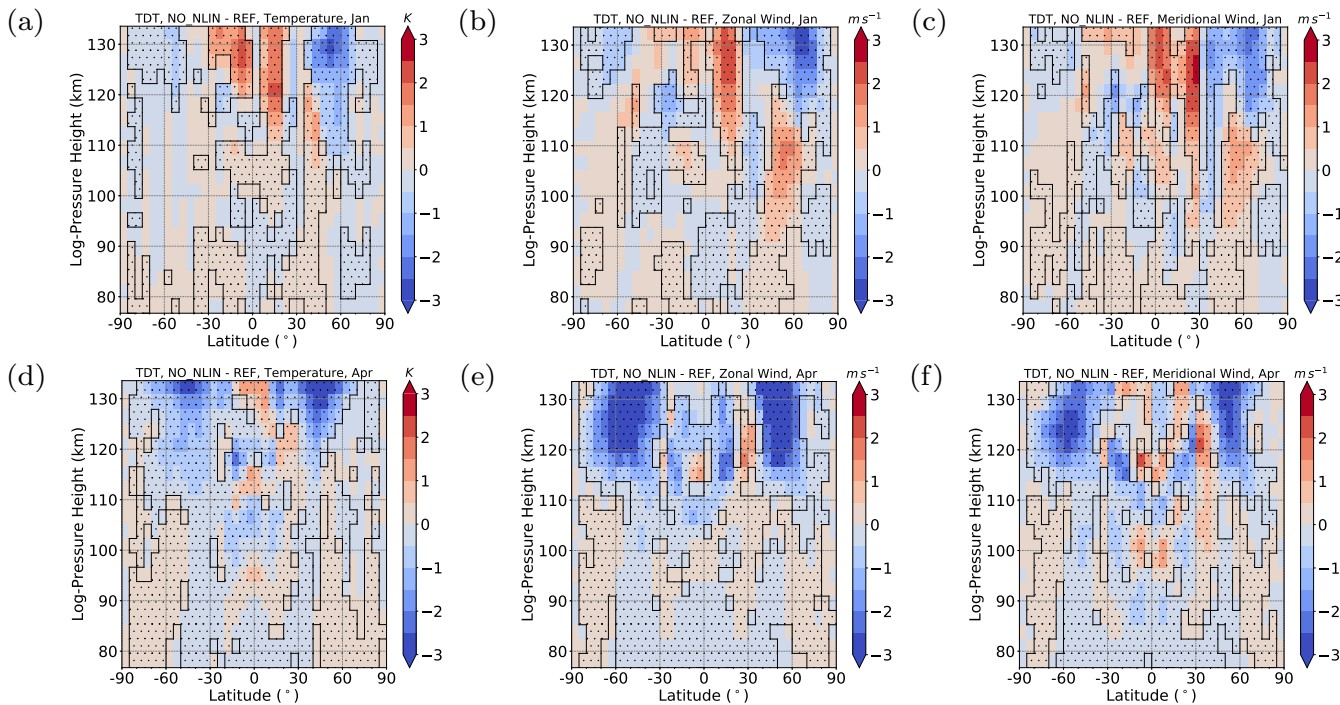

**Figure 10.** Difference of TDT amplitudes between NO_NLIN and REF simulation. Red colors denote larger NO_NLIN simulation amplitudes and blue colors denote larger REF simulation amplitudes. Significant areas ($\alpha < 0.01$) are hatched. From left to right: Temperature, zonal wind, meridional wind. Top: Solstice (January) conditions. Bottom: Equinox (April) conditions.

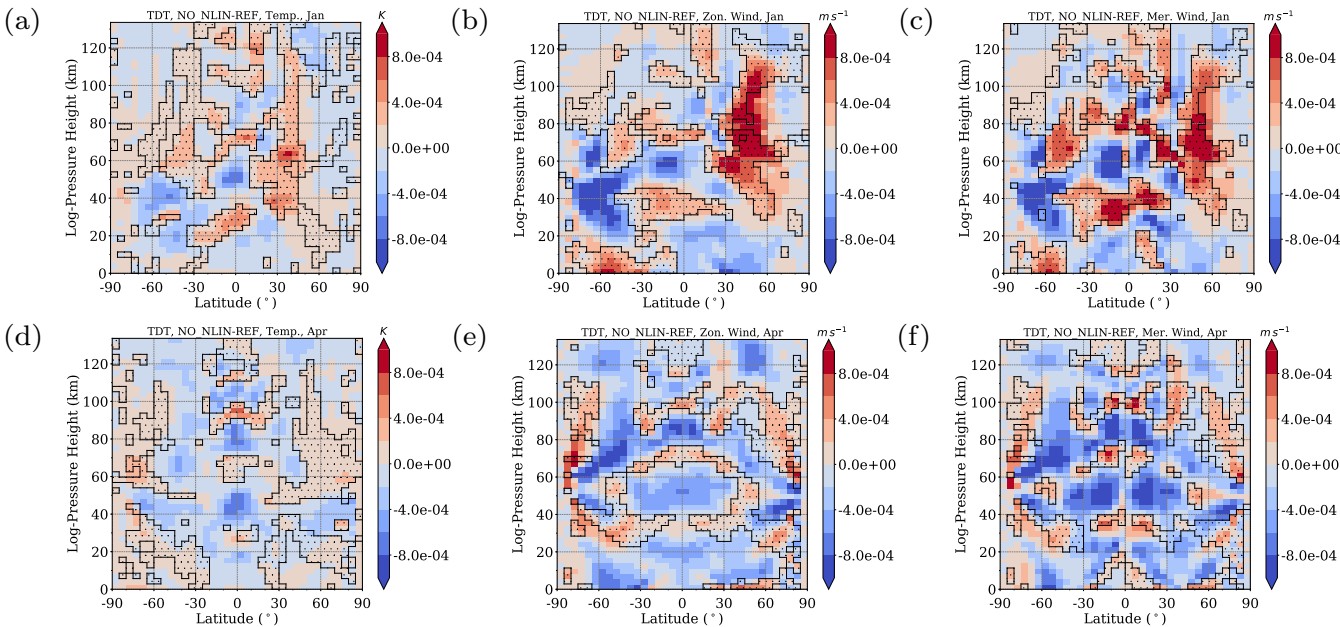

**Figure 11.** Difference of TDT amplitudes between NO_NLIN and REF simulation, scaled by $\exp\{-z(2H)^{-1}\}$. Red colors denote larger NO_NLIN simulation amplitudes and blue colors denote larger REF simulation amplitudes. Areas of destructive interferences ($120° \leq \Delta\Phi \leq 240°$) between NO_NLIN and NO_SOL phases are hatched. From left to right: Temperature, zonal wind, meridional wind. Top: Solstice (January) conditions. Bottom: Equinox (April) conditions.

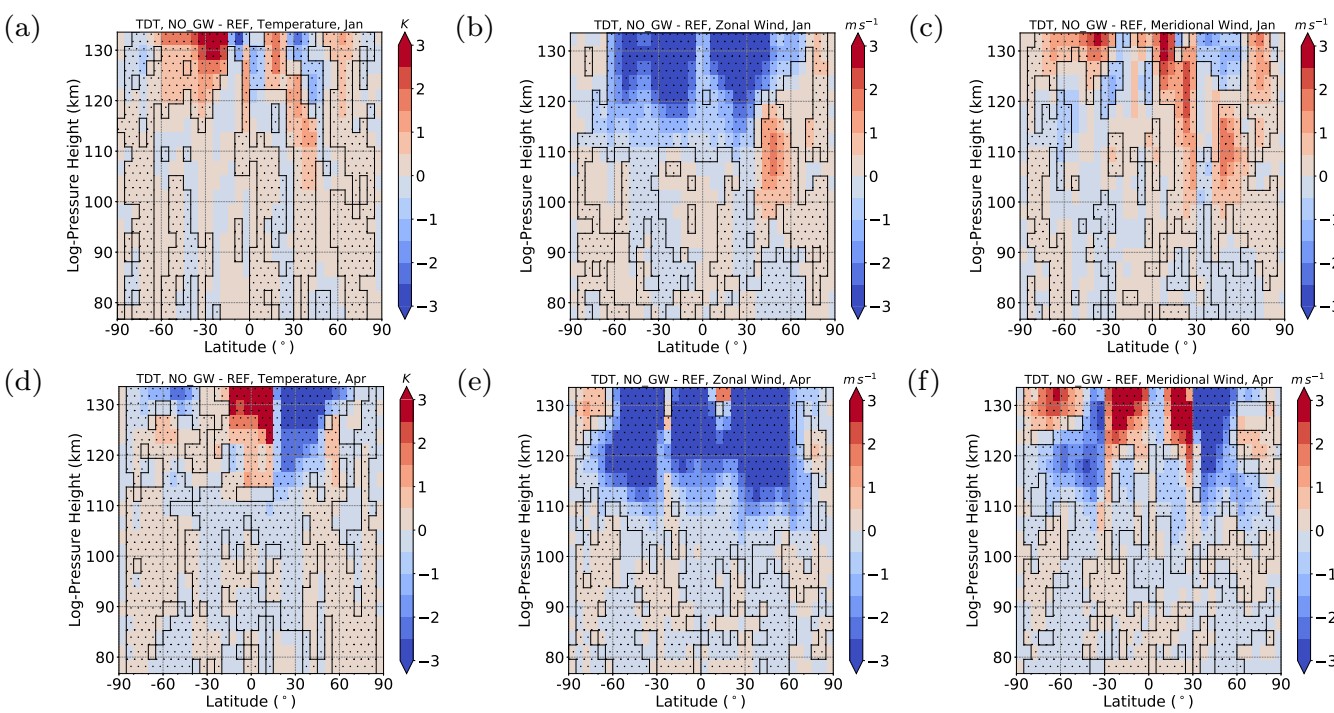

**Figure 12.** As in Fig. 10 but for NO_GW simulation.

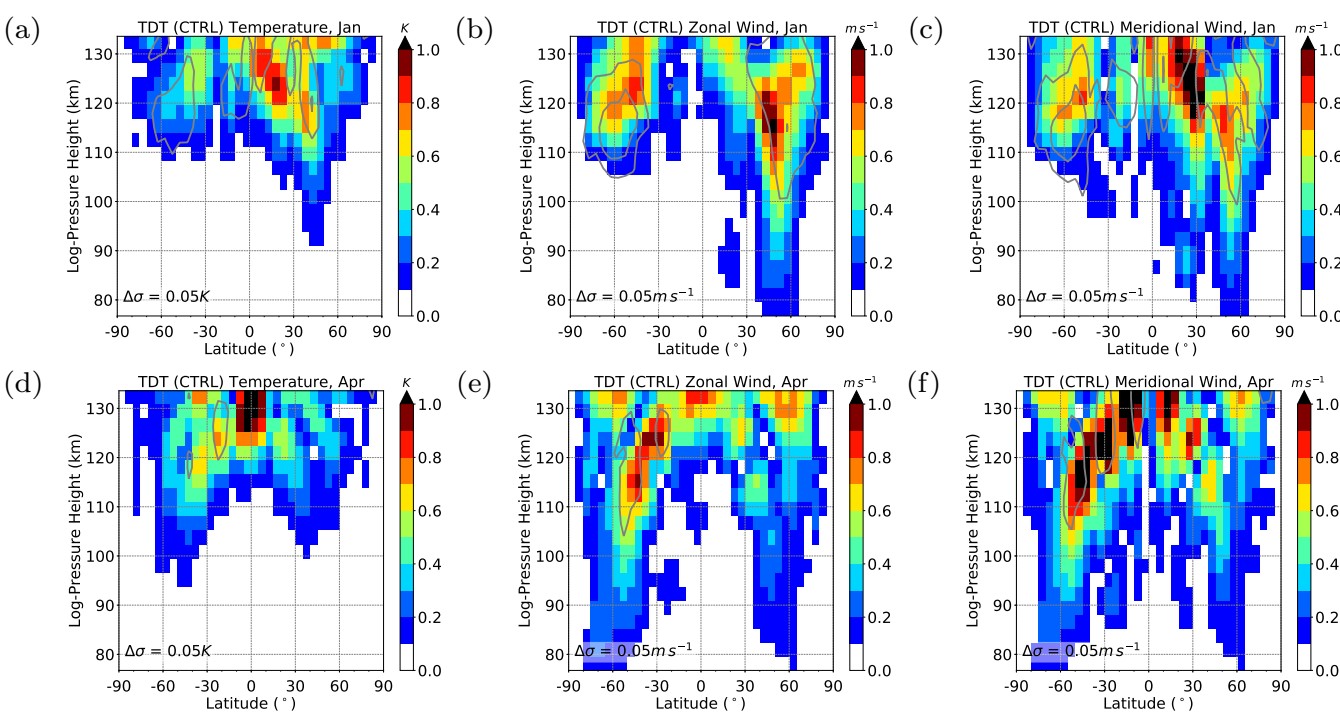

**Figure 13.** As in Fig. 5 but for CTRL simulation. Note that scales are different.