# Peer review of "Forcing Mechanisms of the Terdiurnal Tide"

_Atmospheric Chemistry and Physics, 2018_

## Referee Comment (RC1) · Anonymous Referee #1 · 16 May 2018

**General Comment**

The paper "Forcing Mechanisms of the Terdiurnal Tide" by Lilienthal et al. investigates the forcing of the migrating terdiurnal tide (TDT), a tidal mode that is still not well understood. Several model simulations with the nonlinear Middle and Upper Atmosphere Model (MUAM) are carried out to isolate the effect of different forcing mechanisms: absorption of solar radiation, nonlinear tidal interactions, and the interaction between gravity waves and tides. It is found that direct solar forcing is the dominant process, but also the other processes contribute. Interestingly, the nonlinear forcings can counteract the direct forcing and lead to reduced tidal amplitudes.

Overall, the paper is well written, and it provides very interesting results. Therefore, publication in ACP is recommended after addressing my minor comments.

[Figure]

My main comment is:

From the manuscript, it was not completely clear to me whether removing of forcings would alter the atmospheric background and whether an altered background could have some effect on the TDT in addition to changes in the forcing.

Please find more detailed specific and technical comments below.

**Specific Comments**

1. p.2, l.7/8:
   Miyahara and Forbes (1991) used a modified Lindzen parameterization to investigate the interaction between gravity waves and tides. In this approach gravity wave physics was very simplified. It should be mentioned that more recent simulations show that details of gravity-wave tidal interactions can change if more comprehensive physics is included (for example, Ribstein and Achatz, 2016).

   Ribstein, B., and U. Achatz (2016), The interaction between gravity waves and solar tides in a linear tidal model with a 4-D ray-tracing gravity-wave parameterization, J. Geophys. Res. Space Physics, 121, 8936-8950, doi:10.1002/2016JA022478.

2. p.3, l.9/10: Please explain in more detail:
   Why do you want to avoid the coupling between stationary planetary waves and tides?

3. p.3, l.11:
   Which gravity wave parameterization is used in your simulations?

4. p.3, l.27:
   The statement "The last source might be gravity waves" sounds too weak!

There is evidence for the interaction between gravity waves and tides. For example, on p.3, l.28, it could be mentioned that, indeed, a longitudinal variation of gravity wave activity in the tropical MLT region has been observed that may be caused by an interaction between tides and gravity waves (Trinh et al., 2018, their Fig.4).

Citation: Trinh, Q. T., Ern, M., Doornbos, E., Preusse, P., and Riese, M.: Satellite observations of middle atmosphere–thermosphere vertical coupling by gravity waves, Ann. Geophys., 36, 425-444, https://doi.org/10.5194/angeo-36-425-2018, 2018.

5. p.3, l.34:
   Latent heat release in the troposphere has a zonal wavenumber 3 structure. Do you think that this latent heat release could contribute to the forcing of the migrating TDT as was speculated by Pancheva et al. (2013)?

6. p.4, l.2: Please clarify!
   I think that eliminating each forcing separately is a good approach! However, does eliminating of forcings alter the atmospheric background state? If it does: How do you avoid that changes in the background state cause some variations in the TDT that are then attributed to changes in the forcing mechanisms? Or are variations of the background state negligible compared to the effect of variations in the TDT forcing?

7. p.5, l.19/20:
   Please clarify whether the forcing at all zonal wavenumbers is "switched off" for the CTRL run.

8. p.6 about Figs. 2 and 3:
   Please explain: why are values scaled with the density factor? I think that unscaled values would be more intuitively related to TDT amplitudes in K or m/s.
9. p.9, l.28/29:
   Here you write that "The structure of this remaining tide is not completely irregular indicating that it is possibly not owing to noise."

   Still, noise could be the driver of this tidal structure. Please note that even the numerical noise of a GCM dynamical core can cause "regular oscillations". For example, it has been noted by Rind et al. (2014) that numerical noise can cause QBO-like oscillations in a model.

   Rind, D., J. Jonas, N. K. Balachandran, G. A. Schmidt, and J. Lean (2014), The QBO in two GISS global climate models: 1. Generation of the QBO, J. Geophys. Res. Atmos., 119, 8798-8824, doi:10.1002/2014JD021678.

**Technical Comments**

- p.1, l.17: suggestion for clarification:
  higher wavenumbers → higher wavenumbers / higher frequencies

- p.2, l.19: have been → were

- p.3, l.33: owing to → excited by

- p.5, l.14:
  attributes the thermosphere → takes effect in the thermosphere

- p.5, l.25: ensembles = years ?

- p.7, l.23: is smaller. → is lower.

---

## Author Comment (AC1) · 30 May 2018

Dear anonymous reviewer #1,

on behalf of all co-authors I would like to thank you for your comments and ideas to improve the manuscript. Hereby, I would like to give a short statement on your main comment; a more detailed and comprehensive answer to your review will follow soon.

You raise the question whether removing one forcing will influence the background circulation and therefore also wave propagation conditions and the remaining terdiurnal forcings. As an example, Fig. 1 shows the differences of the zonal mean (background) zonal wind between the NO_SOL and the REF simulation (red and blue colors) where areas of significance are hatched for a 90% confidence interval. For orientation, the REF background zonal wind is added as gray contour lines. It can be seen that significant changes only appear at some spots above 100km altitude. They reach a maximum

[Figure]
* * *
Interactive
comment

of about 1m/s. For comparison, the year-to-year variability of the model (see Fig. 1b in the manuscript) reaches magnitudes of more than 10m/s for the background zonal wind. The variability due to the removed solar forcing is therefore much smaller and we can neglect this effect.

Fig. 2 shows changes in the scaled nonlinear zonal forcing between the NO_SOL and REF simulation (red and blue colors). The REF nonlinear zonal forcing is added as green contour lines for orientation. Differences amount to less than $10^{-3}$m/s/d. In the manuscript, we show the year-to-year variability of this forcing in Fig. 3a which reaches more than $10^{-2}$m/s/d. This also shows, that the internal variability of the model is larger than the effect of removing a terdiurnal forcing term.

Of course, these two figure are only examples. However, note that differences in the background circulation and in nonlinear forcings are even smaller in the NO_GW simulation.

In the next version of the manuscript we will include a paragraph discussing this topic but we do not think that it is necessary to include the figures. These are only for illustration for the reviewer.
* * *
**Zon. Wind Diff. NO_HS-REF, Jan**  $ms^{-1}$

$\Delta_{max}$ = 1.51m/s
$\Delta_{min}$ = -1.22m/s

**Fig. 1.** Differences of January mean background zonal wind (NO_SOL-REF) in colors. Areas of significance are hatched (90% confidence interval). REF background zonal wind as gray contour lines.

**TDT Zon. Wind Acc., Jan $\quad 10^{-3}ms^{-1}d^{-1}$**

**Fig. 2.** Differences of January mean nonlinear zonal forcing (NO_SOL-REF) in colors. REF nonlinear zonal forcing as green contour lines.

---

## Referee Comment (RC2) · Anonymous Referee #2 · 25 Jul 2018

The paper "Forcing Mechanisms of the Terdiurnal Tide" by Lilienthal et al. investigates the nature of the mechanisms which cause terdiurnal tides using a on-linear mechanistic global model, the Middle and Upper Atmosphere Model (MUAM). The approach involves analysing various terms in an ensemble of model runs which use monthly zonal mean temperature fields from ERA-interim reanalysis data to nudge the lower 30 km of the model. The relative role of various terms are diagnosed by doing ensemble runs with specific wave number 3 forcing terms removed. The conceptual framework of the paper is sound but there are a number of issues with the paper that need to be addressed before it can be published in ACP. These include an examination of the solar heating in the model, inclusion of the diurnal and terdiurnal tidal amplitudes which are thought to cause the non-linear interactions and a discussion of the relative amplitudes of these components at the heights at which non-linear effects become important, some discussion of whether the forcing terms result in propagating components

or local forcing (i.e. trapped components) and clarification of the details of the model runs used in this paper. These points are discussed in more detail along with some other less significant points below.

The solar forcing in the this model (discussion on Pg 7, lines 204 to 206) has a similar form (Figure 2) to that published in Smith and Oortland (2001) and Du and Ward (2010) but is over an order of magnitude smaller. The UV heating parameterizations in these models are the same as the one used in the current paper. Furthermore, in the discussion of the terdiurnal amplitudes (pg 8-9, lines 257 to 285) the model amplitudes are generally significantly smaller than those observed. They are also significantly smaller than the amplitudes reported by Du and Ward (2010) in their model run. These smaller amplitudes are consistent with the difference in Solar heating between this model and earlier modelling papers noted above. The authors should investigate the source of these differences and confirm that the heating in the model is correct. Issues with the heating will affect the later sections of the paper so I have not commented on Sections 3.2 and 4.

Some discussion of the amplitudes of the diurnal and semi-diurnal tides in this model should be included. In particular, if non-linear interactions are indicated at a particular height it is important to know what the form of the parent waves is (i.e. the diurnal and semidiurnal tides). It is also possible that although there may be indications of non-linear forcing that this forcing might not result in a propagating tide. In addition to the relationship between periods noted in this paper, there are also relationships between the horizontal and vertical wavelength that should be met for the forcing to result in a propagating component (see Teitelbaum and Vial, JGR, 1991). Consideration of these aspects of the forcing should also be discussed.

The authors provide useful and interesting comparisons between their results and observations. Previous modelling studies are mentioned but there is no explicit comparison between the results of this paper and the previous modelling studies (i.e. latitude/height amplitudes and phases, annual variations, forcing mechanisms). This comparison is needed and would provide the reader with a better idea of how this paper advances the field.

The description of the model runs and the exact nature of the numerical experiment can be improved. The authors note that the lower 30 km of MUAM runs are nudged with monthly mean ERA-Interim reanalyses of zonal mean temperature. An ensemble mean is calculated with the individual runs being driven by results from 2000 to 2010. Runs consists of a 120 day spin-up with not tidal forcing followed by a 90 day run of which the last 30 days are used to investigate the terdiurnal tidal signatures. Apart from Figure 6, only results from January and April are presented. Were there any background winds imposed during the runs? Are the runs used for each month, perpetual runs for that month (i.e. no temporal evolution with the Solar elevation angle remaining constant)? Are the results for the last 30 days stable results (i.e. the model run had achieved some sort of equilibrium) and how was this determined? The authors state that there was no planetary wave forcing at the lower boundary. Was any forcing at the lower boundary included? How was the boundary between the free running part of the model and the nudged portion handled (step function or gradual change). When the tidal forcing started to be included, how was this made compatible with the nudging? Is the tidal forcing turned on gradually? How was the nudging undertaken for the 11 ensemble experiments (page 3) daily zonal means). Is the turning on of the tidal heating a step function?

Additional Comments:

Pg 4 line 113: What is meant by "... remove it in each model time step.". Does this mean that the wave number 3 signature is removed from the model run at each time step at each point in the model?

Pg 4, line 116: Include a reference to justify your decision to ignore non-migrating tides based on the source of nonmigrating tides?

Pg 4, line 123: Please comment on whether ozone is included above 50 km. Is there a

step function in the vertical profile of ozone?

Pg 4, line 128: This should be recombination of O and O3.

Pg 6, line 168: Please explain in more detail how the two gravity wave parameterizations are linked. In most parameterizations the upward flux and dissipation is accounted for. How is the linkage between the upward fluxes between these two parameterizations made? It seems from later results (Figures 2 and 3 and discussion at line 217) that the largest effects are associated with the YiËǦgit parameterization and must have come upward through the stratosphere and mesosphere. This should be discussed.

Pg 6, line 173: Please make explicit that it is the wave number 3 component that is removed in the CTRL run.

Pg 6, line 175: To clarify your procedure, explicitly note that only solstice (January) and equinox (April) conditions are being analysed in this paper apart from Figure 6.

Pg 6, lines 179 - 181: Although it is mentioned earlier in the paper, restating that the ensembles consist of runs using data from each of the years 2000-2010 would make the paper clearer.

Pg 7, lines 215 - 218: The diurnal and semi-diurnal tidal amplitudes are small in the troposphere and the background does not include any planetary waves and is nudged to zonal mean temperatures. The tendency terms here likely do not have much to do with the terdiurnal tide and unless the authors have good reason to include discussion of the dynamics of this region, I suggest it be omitted.

Pg 7, lines 227 - 231: Tidal amplitudes are compared at different heights (90 km for the observations, 100 km for the model). Is there a reason for this?

Pg 8, lines 251 - 256: In Figure 5, the phases are plotted from the ground to 130 km but the amplitude of the terdiurnal tide is only significant about ∼80 km as indicated in Figure 4. The vertical wavelength determinations appear to be associated with this

whole height range. They should only be associated with heights where the amplitude is significant. The wavelength determination should be discussed in more detail.

I have also attached a commented pdf of the paper with some suggestions for improvements to the writing.

Please also note the supplement to this comment:
https://www.atmos-chem-phys-discuss.net/acp-2018-154/acp-2018-154-RC2-supplement.pdf

**Supplement:**

Manuscript prepared for Atmos. Chem. Phys. with version 2014/09/16 7.15 Copernicus papers of the LATEX class copernicus.cls. Date: 9 February 2018

**Forcing Mechanisms of the Terdiurnal Tide**

F. Lilienthal1, Ch. Jacobi1, and Ch. Geißler1

[revised manuscript text omitted]

---

## Author Comment (AC2) · 25 Jul 2018

Dear anonymous reviewer #2,

we would like to thank you very much for your comments and ideas to help improve our manuscript. We will carefully go through the details to prepare a revised version.

In the last sentence of your review you write that a commented PDF is attached. Unfortunately, we could not find the attachment in the interactive discussion. Therefore, we kindly ask you to double check and upload this file again. Thank you!

A detailed response to your review will follow soon.

---

## Author Response (AR1)

**List of relevant changes**

**for the article "Forcing Mechanisms of the Terdiurnal Tide" by F. Lilienthal, Ch. Jacobi and Ch. Geißler**

Text:

- The wavelength relation between DT, SDT and TDT has been described
- A proxy for nonlinear interactions between DT and SDT has been introduced and compared with the terdiurnal component of nonlinear forcing
- The model description has been significantly extended
- The description of phases and vertical wavelengths has been extended
- The discussion section has been significantly extended, especially with respect to earlier model studies
- An outlook for future work is given in the end

Figures and Tables:

- Fig. 4 has been added
- Fig. 12 has been corrected (same figures like in Fig. 10 were included in the earlier version)

**Response to Reviewer#1**

**for the article "Forcing Mechanisms of the Terdiurnal Tide" by F. Lilienthal, Ch. Jacobi and Ch. Geißler**

Again, we would like to thank the anonymous reviewer for his comments to help improve the manuscript. The reviewer's comments are repeated here in italic, our answers are noted below. Quotes of text passages are given with page number (P) and line numbers (L) of the revised manuscript.

My main comment is: From the manuscript, it was not completely clear to me whether removing of forcings would alter the atmospheric background and whether an altered background could have some effect on the TDT in addition to changes in the forcing.

In the following figures, we show the difference of the zonal mean zonal wind for the simulations NO\_SOL-REF, NO\_GW-REF and NO\_NLIN-REF for January and April. Significant changes are hatched. Contour lines show the zonal mean zonal wind in the REF simulation. The changes amount to not more than 2m/s.

Largest differences are obtained in July and reach almost 10m/s in NO\_NLIN, 6.5m/s in NO\_GW and 8m/s in NO\_SOL. These values are still of the same order like the inter-annual variability of the model (see Fig. 1 in the manuscript).

As already mentioned in our short answer on 30 May 2018, we conclude that the influence on the background circulation is relatively small when one of the forcings is removed.

We added the following sentences:

"Note that the background (monthly mean zonal mean) circulation is not significantly altered when TDT forcings are removed (not shown here). Differences amount to not more than the actual standard deviations in the REF simulation (Fig. 1). Therefore, the influence of a removed wavenumber 3 forcing is comparable to the year-to-year variation of the background state and propagation conditions for tides remain similar." (P5L1-P5L4)

**1. p.2, l.7/8:**

Miyahara and Forbes (1991) used a modified Lindzen parameterization to investigate the interaction between gravity waves and tides. In this approach gravity wave physics was very simplified. It should be mentioned that more recent simulations show that details of gravity-wave tidal interactions can change if more comprehensive physics is included (for example, Ribstein and Achatz, 2016).

Thank you for this helpful reference. It is now included in the introduction:

"Another possible excitation source are gravity wave-tidal interactions (e.g., Miyahara and Forbes, 1991; Huang et al., 2007). More recent simulations (Ribstein and Achatz, 2016) show that details of gravity wave-tidal interactions can change if more comprehensive physics is included but their analysis do not include the TDT." (P2L19-21)

**2. p.3, l.9/10: Please explain in more detail:**

Why do you want to avoid the coupling between stationary planetary waves and tides?

We added the following sentence:

"This is important because an additional secondary coupling with planetary waves leads to a more complex situation with a more complicated quantification of the individual forcing effects." (P3L23-25)

**3. p.3, l.11:**

**Which gravity wave parameterization is used in your simulations?**

We shifted a few sentences about the GW parameterization from the end of this section to where the parameterization is first mentioned:

"Gravity waves are calculated by an updated Lindzen-type parameterization (Lindzen, 1981; Jakobs et al., 1986) as described by Fröhlich et al. (2003b) and Jacobi et al. (2006). Due to the fact that this parameterization does not account for ionospheric effects, it is coupled with a modified parameterization after Yiğit et al. (2008), connected via the eddy diffusion coefficient which is calculated in the Lindzen scheme and then transferred to the Yiğit scheme. Gravity waves with phase speeds of 5-30 m s-1 are handled by the Lindzen scheme while the Yiğit scheme is restricted to phase speeds of 35-105 m s-1. This way, the Lindzen-type parameterization affects the stratosphere and mesosphere and the Yiğit parameterization mainly takes effect in the thermosphere. Overlaps between both parameterizations are small and the forcing terms due to gravity waves are summed in the tendency equation of the model. Further parameterizations of solar and infrared radiation as well as several ionospheric effects such as Rayleigh friction, Lorentz force and ion drag are included." (P3L28-P4L3)

**4. p.3, l.27:**

The statement "The last source might be gravity waves" sounds too weak! There is evidence for the interaction between gravity waves and tides. For example, on p.3, l.28, it could be mentioned that, indeed, a longitudinal variation of gravity wave activity in the tropical MLT region has been observed that may be caused by an interaction between tides and gravity waves (Trinh et al., 2018, their Fig.4).

The sentence has been rewritten accordingly:

" The last source included in MUAM are gravity waves. Miyahara and Forbes (1991) have shown that an interaction between gravity waves and the DT can excite a TDT. Trinh et al. (2018) observed a longitudinal variation of gravity wave activity in the tropical MLT region that may also be caused by gravity wave-tidal interaction." (P4L23-26)

**5. p.3, l.34:**

Latent heat release in the troposphere has a zonal wavenumber 3 structure. Do you think that this latent heat release could contribute to the forcing of the migrating TDT as was speculated by Pancheva et al. (2013)?

Of course, latent heat is a possible source of wavenumber 3 tides, namely the TDT. However, we focus on zonal mean dynamics here. Latent heat is strongly dependent on longitude and will also excite nonmigrating tides. This is beyond the scope of this manuscript.

**6. p.4, l.2: Please clarify!**

I think that eliminating each forcing separately is a good approach! However, does eliminating of forcings alter the atmospheric background state? If it does: How do you avoid that changes in the background state cause some variations in the TDT that are then attributed to changes in the forcing mechanisms? Or are variations of the background state negligible compared to the effect of variations in the TDT forcing?

Please see the answer for your main (first) comment.

**7. p.5, l.19/20:**

Please clarify whether the forcing at all zonal wavenumbers is "switched off" for the CTRL run.

The sentence has been rewritten as follows:

"As a control simulation (CTRL), the wavenumber 3 component of the solar, nonlinear and gravity wave forcings are removed simultaneously." (P6L18-19)

**8. p.6 about Figs. 2 and 3:**

*Please explain: why are values scaled with the density factor? I think that unscaled values would be more intuitively related to TDT amplitudes in K or m/s.*

We included the following sentence for clarification:

"All forcing terms are scaled by density (factor  $\exp\{-z/(2H)\}$ ) in order to highlight the region where the forcing originates from. Therefore, the figures show the source region of tidal excitation but they do not provide any information about propagation conditions." (P7L11-13)

Note that reviewer #2 has raised a question concerning the magnitude of the solar heating in comparison to earlier model studies. To answer his question we added figures of the unscaled solar heating. However, we did not include them in our manuscript because we think that the unscaled values do not tell much about the actual forcing region.

9. p.9, l.28/29:

Here you write that "The structure of this remaining tide is not completely irregular indicating that it is possibly not owing to noise."

Still, noise could be the driver of this tidal structure. Please note that even the numerical noise of a GCM dynamical core can cause "regular oscillations". For example, it has been noted by Rind et al. (2014) that numerical noise can cause QBO-like oscillations in a model.

Thank you for these information and the helpful reference. We think that it is better to put the discussion of the CTRL amplitudes into the discussion section and therefore we removed the speculation about noise at this point. In the discussion section we modified the next-to-last paragraph as follows:

"Rind et al. (2014) have noted that numerical noise can produce regular signatures like a quasi-biennual oscillation. Therefore, noise cannot be excluded as a tidal source in the CTRL simulation. Another reasonable TDT source in our model could be originating from the thermospheric parameterizations, which include some nonlinear terms. These sources, however, are likely to be dependent on the used model and it is not likely that the remaining amplitudes in Fig. 13 have a real meteorological meaning." (P13L13-17)

**Technical Comments**

• *p.1, l.17: suggestion for clarification: higher wavenumbers*  $\rightarrow$  *higher wavenumbers / higher frequencies*

- *p.2, l.19: have been*  $\rightarrow$  were
- *p.3, l.33: owing to*  $\rightarrow$  *excited by*
- *p.5, l.14: attributes the thermosphere*  $\rightarrow$  *takes effect in the thermosphere*
- *p.5*, *l.25*: *ensembles* = *years* ?
- p.7, l.23: is smaller.  $\rightarrow$  is lower.

All your technical comments have been addressed, accordingly.

**Response to Reviewer#2**

**for the article "Forcing Mechanisms of the Terdiurnal Tide" by F. Lilienthal, Ch. Jacobi and Ch. Geißler**

Again, we would like to thank the anonymous reviewer for his comments to help improve the manuscript. The reviewer's comments are repeated here in italic, our answers are noted below. Quotes of text passages are given with page number (P) and line numbers (L) of the revised manuscript.

**General concerns:**

The solar forcing in the this model (discussion on Pg 7, lines 204 to 206) has a similar form (Figure 2) to that published in Smith and Ortland (2001) and Du and Ward (2010) but is over an order of magnitude smaller. The UV heating parameterizations in these models are the same as the one used in the current paper. Furthermore, in the discussion of the terdiurnal amplitudes (pg 8-9, lines 257 to 285) the model amplitudes are generally significantly smaller than those observed. They are also significantly smaller than the amplitudes reported by Du and Ward (2010) in their model run. These smaller amplitudes are consistent with the difference in Solar heating between this model and earlier modelling papers noted above. The authors should investigate the source of these differences and confirm that the heating in the model is correct. Issues with the heating will affect the later sections of the paper so I have not commented on Sections 3.2 and 4.

Please note that the solar heating rates presented in Fig. 2 are scaled by density, i.e. they are multiplied by the factor  $\exp\{-z/(2H)\}$ . The unscaled solar heating has a magnitude that is comparable to those presented by Smith and Ortland (2001) and Du and Ward (2010). The following figures show the unscaled zonal mean heating rates (left), TDT component of unscaled solar heating rates (middle) and the decomposition of daily solar heating rates at 2.5°N/50km into its daily mean and tidal components which are centered around the mean value (right). We hope to convince you that the solar radiation parameterization in the model works correctly.

In order to avoid confusion, we mention the scaling at several additional text passages to remind the reader, e.g.:

"Figures 2 and 3 show the terdiurnal component of all forcing terms that our analysis takes into account, namely solar forcing, nonlinear forcing and forcing due to gravity wave-tide interactions. All forcing terms are scaled by density (factor  $\exp\{-z/(2H)\}$ ) in order to highlight the region where the forcing originates from. Therefore, the figures show the source region of tidal excitation but they do not provide any information about propagation conditions." (P7L10-13)

---

## Referee Report (RR1)

[referee-annotated manuscript omitted]

---

## Author Response (AR2)

**Response to Reviewer#2**

**for the article "Forcing Mechanisms of the Terdiurnal Tide"**
**by F. Lilienthal, Ch. Jacobi and Ch. Geißler**

Again, we would like to thank the anonymous reviewer for his comments to help improve the manuscript. The reviewer's comments are repeated here in italic, our answers are noted below. Quotes of text passages are given with page number (P) and line numbers (L) of the revised manuscript.

*In the paper under review, the authors examine the forcing for steady state conditions (i.e. without the variability inherent in a full GCM - this situation is similar to the longer time-scale situation discussed by Du and Ward (2010)), demonstrate that there are regions where non-linear forcing terms are non-zero and show using suitably selected runs that the tidal amplitudes vary in a way that is consistent with non-linear interactions playing a role in the terdiurnal tidal amplitudes. They thereby forward our understanding of aspects of this tide. However, the authors do not directly show that the forcing terms generate propagating terdiurnal tides as opposed to simply resulting in a local forcing nor do they distinguish whether the modes which might be forced non-linearly are the same as those which are thermally forced. It seems that the Hough mode analysis suggested by Du and Ward could contribute to clarifying what is going on with MUAM also (I am not suggesting that this be attempted as a revision to this paper). I do suggest that authors, provide more nuanced conclusions (page 13) to their paper, point to their contribution to studying this question and note that the issue of non-linear interactions for the terdiurnal tide is still an open question which warrants further study.*

> The reviewer raises an interesting concern about tidal modes of the forcing terms in relation to propagating tidal modes. We feel that this issue is beyond the scope of our manuscript, but to nevertheless address this point we added a paragraph in the discussion concerning a possible Hough decomposition for future analysis.

> "In order to investigate the different generation mechanisms of the TDT we present their respective source regions. In addition to the methods used by, e.g. Akmaev (2001); Smith and Ortland (2001); Huang et al. (2007) or Du and Ward (2010), who focus on direct solar heating and nonlinear interactions between tides only, we also consider gravity wave-tide interactions as suggested by, e.g. Miyahara and Forbes (1991) and Huang et al. (2007). To summarize, the solar forcing is dominant in the troposphere and stratosphere, nonlinear interactions are present in the mesosphere and the forcing due to gravity waves is only significant in the zonal component above the mesopause. This analysis, however, only allows insight into the local terdiurnal forcing which does not necessarily result in a propagating tide. As suggested by Du and Ward (2010), a Hough mode decomposition of the forcing terms and of the actual tide could shed more light into this issue. Here, a different approach has been applied to analyze the propagating tides due to different forcing mechanisms. Similar to Akmaev (2001) and Smith and Ortland (2001), we perform further model simulation where possible forcing mechanisms are switched off individually." (P12L20-29)

*The new discussion on the criteria (page 3) associated with the generation of terdiurnal tides is not complete. Since this tide is a global mode with specific parameters associated with each mode, an additional criteria is the extent to which the forcing conditions match the parameters associated with the tidal mode. If there is a significant mis-match then there will be local forcing but no propagating modes will be excited. This aspect of the forcing should be included as part of this discussion.*

Similar to the answer of the previous comment, we added a paragraph in the introduction of our manuscript that explains the advantages of a Hough mode decomposition.

"They [Du and Ward (2010)] concluded that nonlinear interactions are unlikely to be the source of the migrating TDT and that solar heating is the major source. However, Du and Ward (2010) do not exclude the possibility of nonlinear interactions. They suggest a Hough mode decomposition of the TDT, similar to the analysis of Smith and Ortland (2001). This procedure allows the conclusion to which degree a local forcing actually results in a propagating tidal mode." (P2L34-P3L3)

*I am still curious as to how the gravity wave scheme in the model is implemented (page 3) and suggest a little more explanation is still needed. The authors state that the two schemes are separated in terms of the wavenumber range that each is associated with and suggest that this results in each scheme affecting different height ranges. However, won't the wind filtering in the stratosphere and mesosphere affect phase speeds outside the +/- 35 m/s boundary between the two schemes so that both are active throughout the full altitude range? Please comment on this.*

The referee is right, both GW schemes are active throughout the whole altitude range. However, according to linear theory, gravity waves with phase speed c and amplitude A start to break when the wave maximum reaches its critical line (c+A=u). Therefore, waves with large phase speeds need to have a larger amplitude than waves with small phase speeds to transfer energy to the mean flow. Therefore, they need to propagate to higher altitudes (>100km) where the amplitude is larger to reach the critical line. The difference of affected height regions by the Lindzen-type and Yiğit scheme is therefore mainly due to the different phase speed range chosen.

Positive phase speeds are either filtered below the zonal wind jet maximum (e.g. c<80m/s) or they propagate up to the thermosphere (c>80m/s) but they do not strongly contribute to the wind reversal in the mesosphere.

The following figures present the GW drag of both parameterizations in MUAM, the linear Lindzen-type scheme (left) and the Yiğit scheme (right), with model boundary conditions for January 2000.

[Figure]

[Figure]

We modified the text as follows:

"Gravity waves with phase speeds of 5 to 30 m/s are handled by the linear Lindzen-type scheme while the Yiğit scheme is restricted to phase speeds of 35 to 105 m/s. Therefore, the intrinsic phase speeds of the waves in the Yiğit scheme are larger than those in the Lindzen-type scheme, so that they reach their breaking levels at higher altitudes where the amplitude is larger. As a result, the Lindzen-type parameterization essentially affects the stratosphere and mesosphere and the Yiğit parameterization mainly takes effect in the thermosphere. Overlaps between both parameterizations are small..." (P4L2-7)

*The exp(-z/2H) weighting used in this paper is not a "density weighting" but a term associated with conservation of wave energy which normalizes the wave growth with height due to the decrease in density. I think this is an appropriate way to compare different forcings but suggest you use a different term to describe it.*

We avoided the term "density weighting" and called it now "scaling by growth rate", introduced on P7L20-22:

"All forcing terms are scaled by $\exp\{-z/(2H)\}$. This factor is associated with the conservation of wave energy which normalizes the wave growth with height due to the decrease in density."

*Pg 5, Line 13: I apologize for the typo in my last review. The species which recombine are O and O2 to produce O3 as the authors show in the reactions that they list in their response. O3 is the resulting species not the molecule which recombines. The text should be changed to "recombination of O and O2".*

Thank you again for pointing this out. We changed it accordingly (P5L19).

*Pg 8, line 1: "subharmonics" should be changed to "harmonics". Subharmonics are modes with frequencies lower than the parent waves. Harmonics are modes with frequencies higher than the parent waves.*

"Subharmonics" has been changed to "harmonics" (P1L12,P1L16,P8L11).

*Pg 8, line 4-5: I don't understand why the scaled amplitudes are omitted from the figures since they are of a higher order of magnitude. If they are of a higher order of magnitude, doesn't that mean that the interaction is more significant? Please comment.*

We wish to thank the reviewer for his comment as it helped to find a mistake in our analysis. In the previous version, DT and SDT amplitudes have been scaled, first, and then multiplied with each other. This includes one more scaling than necessary, i.e. the factor is $\exp\{-z/H\}$ instead of $\exp\{-z/(2H)\}$. The correct way is to multiply the DT and SDT amplitudes first and then scaling the result. This has been corrected and the respective figures now extend from the surface to 130km, similar to the other amplitude plots. We changed the selection of the subfigures to show the nonlinear coupling in all three parameters, temperature, zonal and meridional wind. The result shows very good agreement with the nonlinear forcing terms.

"To test this relation between the different harmonics, the product of DT and SDT amplitudes serves as a proxy for the terdiurnal nonlinear forcing. Due to the fact that the forcing terms in Figure 3 are scaled by the growth rate of the amplitudes, we also scaled the product of DT and SDT amplitudes to show the source region of the possible interaction. As an example, Fig. 4

shows the results for temperature (a), zonal wind (b) and meridional wind amplitudes (c) during January." (P8,L10-14)

*I have a number of suggestions for improvements to the wording which I will forward separately as comments on the revised manuscript.*

We greatly appreciate the help of the reviewer concerning grammar and wording and apologize for the number of language issues.

**List of relevant changes**

**for the article "Forcing Mechanisms of the Terdiurnal Tide"**
**by F. Lilienthal, Ch. Jacobi and Ch. Geißler**

Text:
- A paragraph with respect to tidal modes and Hough mode decomposition has been added in the introduction and discussion sections of the manuscript

Figures and Tables:
- Fig. 4 has been changed and a mistake in the analysis has been corrected (for details see response to Referee #2)
- All figures have been slightly adjusted, now showing the whole latitudinal range between 90°S to 90°N and information about intervals of the standard deviations have been added in each panel. The data remain unchanged (except for Fig. 4).

**Forcing Mechanisms of the Terdiurnal Tide**

Friederike Lilienthal[1], Christoph Jacobi[1], and Christoph Geißler[1]

[1]Institute for Meteorology, Universität Leipzig, Stephanstr. 3, 04103 Leipzig, Germany

**Correspondence:** F. Lilienthal (friederike.lilienthal@uni-leipzig.de)

**Abstract.** Using a nonlinear mechanistic global circulation model we analyze the migrating terdiurnal tide in the middle atmosphere with respect to its possible forcing mechanisms, i.e. the absorption of solar radiation in the water vapor and ozone band, nonlinear tidal interactions, and gravity wave-tide interactions. In comparison to the forcing mechanisms of diurnal and semidiurnal tides, these terdiurnal forcings are less well understood and there are contradictory opinions about their respective relevance. In our simulations we remove the wavenumber 3 pattern for each forcing individually and analyze the remaining tidal wind and temperature fields. We find that the direct solar forcing is dominant and explains most of the migrating terdiurnal tide's amplitude. Nonlinear interactions due to other tides or gravity waves are most important during local winter. Further analyses show that the nonlinear forcings are locally counteracting the solar forcing due to destructive interferences. Therefore, tidal amplitudes can become even larger for simulations with removed nonlinear forcings.

**1 Introduction**

Atmospheric waves such as solar tides play a crucial role in the dynamics of the mesosphere/lower thermosphere (MLT) region. Tides are global-scale oscillations with periods of a solar day (24 h) or its  harmonics (12 h, 8 h, etc.). They are mainly the result of absorption of solar radiation in the water vapor (troposphere) and ozone (stratosphere) region. Tidal amplitudes grow with increasing height due to the decrease of density and conservation of energy (e.g., Chapman and Lindzen, 1970; Andrews et al., 1987). In the MLT, tides can reach wind amplitudes comparable to the magnitude of the horizontal mean wind.

Due to the fact that diurnal tides (DTs) and semidiurnal tides (SDTs) usually have larger amplitudes than the harmonics of higher wavenumbers/higher frequencies, they have attracted more attention in the past and are therefore relatively well understood. However, there are observations of terdiurnal tides (TDTs) showing local amplitudes comparable to those of DTs during some months of the year (Cevolani and Bonelli, 1985; Reddi et al., 1993; Thayaparan, 1997; Younger et al., 2002; Jacobi, 2012). Observations using midlatitude radar measurements show large TDT amplitudes in autumn and early winter (Beldon et al., 2006; Jacobi, 2012). Namboothiri et al. (2004) also obtained slightly larger amplitudes in winter than in summer while Thayaparan (1997) and Jacobi (2012) additionally emphasize the occurrence of TDTs during spring.

Satellite observations have been used to analyze the TDT on a global scale (Smith, 2000; Moudden and Forbes, 2013; Pancheva et al., 2013; Yue et al., 2013). Yue et al. (2013) presented TDT wind amplitudes from the Thermosphere Ionosphere Mesosphere Energetics and Dynamics (TIMED) Doppler Interferometer (TIDI) of more than $16\,\mathrm{m\,s^{-1}}$ at $50°$ N/S above $100\,\mathrm{km}$ with an

additional peak in the meridional component at about $82\,\text{km}$ between 10 and $20°$ N. They identified the first symmetric (3,3) mode (peaking at $8\,\text{K}$ above the equator and at midlatitudes), using temperatures from Sounding of the Atmosphere using Broadband Emission Radiometry (SABER). At an altitude of $90\,\text{km}$, Moudden and Forbes (2013) found the largest amplitudes above the equator during equinoxes ($6-8\,\text{K}$), and also at $60°$ N during May ($7\,\text{K}$) and at $60°$ S during during October ($5\,\text{
[revised manuscript text omitted]